

# Numerical Simulations of the 2004 Indian Ocean Tsunami Deposits Thicknesses and Emplacements

Syamsidik[1,2], Musa Al'ala[1], Hermann M. Fritz[3], Mirza Fahmi[1,4], Teuku Mudi Hafli[1]

[1]Tsunami and Disaster Mitigation Research Center (TDMRC), Syiah Kuala University, Jl. Prof. Dr. Ibrahim Hasan, Gp. Pie,
Banda Aceh, 23233, Indonesia
[2]Civil Engineering Department, Faculty of Engineering, Syiah Kuala University, Jl. Syeh Abdurrauf No. 7, Banda Aceh,
23111, Indonesia
[3]School of Civil Engineering and Environmental Engineering, Georgia Institute of Technology, 790 Atlantic Drive, Atlanta,
GA 30332, USA
[4]Civil Engineering Department, Almuslim University, Bireuen, Indonesia

*Correspondence to*: Syamsidik (syamsidik@tdmrc.org, syamsidik@unsyiah.ac.id )

**Abstract.** After more than a decade of recurring tsunamis, identification of tsunami deposits, a part of hazard characterization, still remains a challenging task not fully understood. The lack of sufficient monitoring equipment and rare tsunami frequency are among the primary obstacles that limit our fundamental understanding of sediment transport mechanisms during a tsunami. The use of numerical simulations to study tsunami-induced sediment transport was rare in Indonesia until the 2004 Indian Ocean tsunami. This study aims to couple two hydrodynamic numerical models in order to reproduce tsunami-induced sediment deposits, i.e., their locations and thicknesses. Numerical simulations were performed using the Cornell Multi-Grid Coupled Tsunami Model (COMCOT) and Delft3D. This study reconstructed tsunami wave propagation from its source using COMCOT, which was later combined with Delft3D to map the location of the tsunami deposits and calculate their thicknesses. Two Dimensional-Horizontal (2DH) models were used as part of both simulation packages. Lhoong, in the Aceh Besar District, located approximately 60 km southwest of Banda Aceh, was selected as the study area. Field data collected in 2015 and 2016 validated the forward modeling techniques adopted in this study. However, agreements between numerical simulations and field observations were more robust using data collected in 2005, i.e., just months after the tsunami (Jaffe et al., 2006). We conducted pit (trench) tests at select locations to obtain tsunami deposit thickness and grain size distributions. The resulting numerical simulations are useful when estimating the locations and the thicknesses of the tsunami deposits. The agreement between the field data and the numerical simulations is reasonable despite a trend that overestimates the field observations.
.

## 1 Introduction

In recent decades, sediment transport dynamics due to extremely long waves, such as tsunamis, has increasingly caught the interest of researchers, especially from countries in the direct path of these phenomena. Studies of tsunami impact on coastal morphology, involving a comparison with actual field data, are rare for previous tsunami events. This limits our ability to understand the physical processes occurring during erosion and sediment deposition. The 2004 Indian Ocean tsunami, triggered by a magnitude $M_w$ 9.1 earthquake, dramatically changed coastal areas around Aceh of Indonesia (Borrero et al.,


2006). Erosion processes and its effects were quite profound and severe along most of Aceh's west coast (Syamsidik et al., 2015). Severe erosion associated with the 2004 tsunami actually created a new island (Al'ala et al., 2015). Tsunami waves transported large volumes of sediment of several hundred meters at certain locations, resulting in a large-scale coastline recession. Large shear stresses generated by the long waves overpowered the critical shear stress of bedload materials

throughout the coastal zone. However, full investigation into the physical processes of long wave shear stresses during the 2004 tsunami did not occur. Owing to the lack of sufficient coastal monitoring equipment in tsunami prone areas and the rare occurrence of tsunamis, insights into sediment transport mechanisms during tsunami events remain poorly understood. Overland flow velocities extracted from spontaneous eyewitness videos recorded during the 2004 Indian Ocean tsunami are limited to a few inland locations in Banda Aceh (Fritz et al., 2006). The coupling of tsunami hydrodynamics and sediment

transport warrants future studies in order to improve our fundamental understanding of tsunami processes. Before the 2004 tsunami, paleo-tsunami studies were quite rare in this region (Andrade et al., 2014).

Two major factors, i.e., episodic land subsidence due to tectonic movement and massive sediment transport during tsunami propagation, play various roles in modifying coastal morphology (Moore et al., 2006). Several studies conducted in this

region include the investigation of tsunami deposits at Khao Lak in Thailand (Dawson and Stewart, 2007; Jankaew et al., 2008) and Meulaboh in the Aceh Province (Monecke et al., 2008). Another study reported the presence of tsunami sediment deposits in a coastal cave, which is less than 5 km from our study area in Lhoong of Aceh Besar (Rubin et al., 2017). Both studies revealed that information from tsunami deposits is capable of improving our understanding of the physical processes involved during the transport of sediment. The recurrence rate of tsunamis throughout this region, information critical for

hazard assessment, is inferable from events recorded in the tsunami deposits. Sediment transport due to tsunami waves has certain characteristic properties and hydrodynamic regimes during both the tsunami run-up and backwash. During transport, material that is mobile depends on grain size and density, which vary greatly because of the erratic nature of tsunami sediment transport. Very few studies investigated the energy involved during sediment transport produced by the 2004 tsunami in Aceh. Furthermore, only a few of these published studies confirmed the thickness of the sediment deposits after

the tsunami via geological surveys. This is the first study in northern Sumatra which applies numerical simulations in order to estimate the locations of tsunami deposits prior to the field surveys. Earlier investigations concentrated on forward tsunami modeling, such as in Kuala Meurisi (Apotsos et al., 2011), Lhok-Nga (Gusman et al., 2012; Li et al., 2012), and Ulee Lheue Bay (Syamsidik et al., 2017). This research aims to couple two hydrodynamic numerical models in order to reproduce the spatiality of the 2004 tsunami sediment deposits, i.e., their locations and thicknesses. We analyze the possible

sources of large shear stresses produced by the tsunami which transported sediment grains of various sizes. Several studies observed that backwash processes, associated with tsunami waves, deposited large volumes of sediments offshore (Jiang, 2015). Sediment properties, such as grain size, correlate poorly with the thicknesses of tsunami sediment deposits (Cheng and Weiss, 2013). Another poorly characterized physical process related to the tsunami is the location and extent of inland sediment deposition, which differs from the extent of tsunami inundation (Goto et al., 2012).

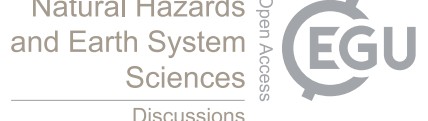

We combine both numerical simulations and field surveys to investigate sediment transport induced by the tsunami at Lhoong of Aceh Besar. We also conducted topographic surveys and sediment deposit pit tests in the vicinity. Studies rarely focus on changes along the shoreline due to the impacts of tsunami waves and subsequent erosion and deposition.

Morphologic changes to the shoreline and sediment thickness are attributable to the amount of energy involved in tsunami propagation and inundation. This study estimates the energy required to transport sediments via a tsunami wave. The trajectory of the 2004 Indian Ocean tsunami was numerically simulated using the Cornell Multi-grid Coupled Tsunami (COMCOT) model, from its rupture area to the shoreline, and we applied Delft3D-FLOW to simulate sediment transport processes by inland flooding in the study area. The combination of these two models provides an ideal re-creation of the

2004 tsunami scenario, since COMCOT does not include sediment transport and the original version of the Delft3D model does not have the option of multi-fault scenarios to generate tsunami waves (Syamsidik and Istiyanto, 2013). The numerical model is described briefly in Section 3, and the model results are presented in Section 4. Field observations are also described in Section 4. The observations and model results are discussed in Section 5, and conclusions are presented in Section 6.

## 2 Study Area

To carry out the objectives of this study, we selected an area where human intervention has remained minimal during the decade following the 2004 tsunami in order to obtain well-preserved tsunami deposits. The study area is located at Lhoong in the Aceh Besar District, Indonesia (Fig. 1). The 2004 Indian Ocean tsunami destroyed a wide swath of coastal area, including the Aceh Besar District. The study area is situated on the west coast of Sumatra approximately 40 km south of

Banda Aceh, the city most devastated by the tsunami. Several prior studies have been conducted in the general study area (Jaffe et al., 2006; Rubin et al., 2017). Coastal mountains, representing extension of the Bukit Barisan Mountains, surround the study area. We selected four specific areas for detailed sediment transport investigations. The selected areas are shown in red boxes in Fig. 1, namely, Birek, Pasie Janeng, Jantang, and Saney. Among these locations, only Jantang has a wide coastal plain, and the other three sites have more rugged terrain. Birek and Pasie Janeng have V-shaped coastal areas

characterized by steep hill slopes 100 to 300 m from their coastlines. Saney headland features a narrow plain sandwiched between coastline to both the north and south.

The 2004 tsunami heavily impacted the Lhoong region, leaving a number of areas conserved after the tsunami because of their relative isolation from major human activities. After more than a decade, it is difficult to find a suitable location to

investigate tsunami deposits because many areas have been re-built, modifying the deposits during reconstruction processes. Birek, Pasie Janeng, and Saney are the sites that conserve the best sediment deposits because of the absence of anthropogenic influences that might disturb the tsunami deposits. Furthermore, researchers conducted a field survey of



Jantang 3 months after the tsunami (Peters and Jaffe, 2006; Jaffe et al., 2006). Aerial photographs of the study area are shown in Fig. 2. Birek is surrounded by a range of hills covered by dense forests that regrew rapidly after the 2004 tsunami. The characteristics of Pasie Janeng are similar to Birek. Jantang (Fig. 2(c)), however, has a wider flat area that allowed the tsunami to propagate farther inland compared with Birek and Pasie Janeng. Since the 2004 tsunami, however, Jantang has

5   recovered, new settlements have been established, and it is actively used for agricultural purposes.

## 3 Methods

### 3.1. Numerical Simulations

During the simulation process, we combined two numerical models. COMCOT simulated the formation and propagation of tsunami waves. A series of laboratory experiments and prior tsunami events have validated the use of the COMCOT model

10  (Liu et al., 1995). COMCOT was successfully implemented to investigate historical tsunami events, such as the 1992 Flores Island tsunami (Liu et al., 1995) and the 2004 Indian Ocean tsunami (Wang and Liu, 2006a; Wang and Liu, 2006b; Wang and Liu, 2007; Syamsidik et al., 2015). Using the incident tsunami wave heights generated by COMCOT, Delft3D-FLOW models sediment transport from the ocean to the inundation zone, as well as the reverse flow. Both models coupled the hydrodynamic and morphodynamic models. The hydrodynamic model has been validated comparatively with numerical and

15  laboratory data following several tsunami benchmark standards (Apotsos et al., 2010). For the hydrodynamic model, Delft3D uses a finite difference scheme on a three-dimensional grid, with the non-linear shallow water equations (NSWEs). The linear shallow water equations used in COMCOT in spherical coordinate system are as follows:

$$\frac{\partial \eta}{\partial t} + \frac{1}{R\cos\varphi}\left\{\frac{\partial P}{\partial \psi} + \frac{\partial}{\partial \varphi}(\cos\varphi Q)\right\} = -\frac{\partial h}{\partial t}, \tag{1}$$

$$\frac{\partial P}{\partial t} + \frac{gh}{R\cos\varphi}\frac{\partial \eta}{\partial \psi} - fQ = 0, \tag{2}$$

$$\frac{\partial Q}{\partial t} + \frac{gh}{R}\frac{\partial h}{\partial \varphi} + fP = 0, \tag{3}$$

The equations governing non-linear shallow water behavior in COMCOT for a spherical coordinate system can be expressed as follows:

$$\frac{\partial \eta}{\partial t} + \frac{1}{R\cos\phi}\left\{\frac{\partial P}{\partial \psi} + \frac{\partial}{\partial \phi}(\cos\phi Q)\right\} = -\frac{\partial h}{\partial t}, \tag{4}$$

$$\frac{\partial P}{\partial t} + \frac{1}{R\cos\phi}\frac{\partial}{\partial \psi}\left\{\frac{P^2}{H}\right\} + \frac{1}{R}\frac{\partial}{\partial \phi}\left\{\frac{PQ}{H}\right\} + \frac{gH}{R\cos\phi}\frac{\partial \eta}{\partial \psi} - fQ + F_x = 0 \tag{5}$$





$$\frac{\partial Q}{\partial t} + \frac{1}{R\cos\phi}\frac{\partial}{\partial \psi}\left\{\frac{PQ}{H}\right\} + \frac{1}{R}\frac{\partial}{\partial \phi}\left\{\frac{Q^2}{H}\right\} + \frac{gH}{R}\frac{\partial \eta}{\partial \phi} + fP + F_y = 0,\tag{6}$$

$$f = 2\Omega\sin\phi,\tag{7}$$

$$F_x = \frac{gn^2}{H^{7/3}}P\left(P^2 + Q^2\right)^{1/2}\tag{8}$$

$$F_y = \frac{gn^2}{H^{7/3}}Q\left(P^2 + Q^2\right)^{1/2}\tag{9}$$

$$H = \eta + h.\tag{10}$$

Meanwhile, for Cartesian coordinate system, COMCOT applies the following equations for linear shallow water equations.

$$\frac{\partial \eta}{\partial t} + \left\{\frac{\partial P}{\partial x} + \frac{\partial Q}{\partial y}\right\} = -\frac{\partial h}{\partial t},\tag{11}$$

$$\frac{\partial P}{\partial t} + gh\frac{\partial \eta}{\partial x} - fQ = 0,\tag{12}$$

$$\frac{\partial Q}{\partial t} + gh\frac{\partial h}{\partial y} + fP = 0.\tag{13}$$

For linear shallow water equations, the following formulae were applied in COMCOT.

$$\frac{\partial \eta}{\partial t} + \left\{\frac{\partial P}{\partial x} + \frac{\partial Q}{\partial y}\right\} = -\frac{\partial h}{\partial t},\tag{14}$$

$$\frac{\partial P}{\partial t} + \frac{\partial}{\partial x}\left\{\frac{P^2}{H}\right\} + \frac{\partial}{\partial y}\left\{\frac{PQ}{H}\right\} + gH\frac{\partial \eta}{\partial x} + F_x = 0,\tag{15}$$

$$\frac{\partial Q}{\partial t} + \frac{\partial}{\partial x}\left\{\frac{PQ}{H}\right\} + \frac{\partial}{\partial y}\left\{\frac{Q^2}{H}\right\} + gH\frac{\partial \eta}{\partial y} + F_y = 0,\tag{16}$$

where $\eta$ is the water surface elevation, $(P, Q)$ gives the volume flux in the $X$ (west–east) direction and in the $Y$ (north–south) direction, respectively, $(\varphi, \psi)$ is the latitude and longitude, $R$ is the Earth's radius, $g$ is gravitational acceleration, and $h$ is water depth. The $-\partial h/\partial t$ expression reflects the effect of transient seafloor motion, $f$ represents the Coriolis force coefficient due to the Earth's rotation, $\Omega$ is the Earth's rotation rate, $H$ is the total water depth, $F_x$ and $F_y$ represent bottom friction in the $\psi$ and $\varphi$ direction, respectively, and $n$ is Manning's roughness coefficient. Since Delft3D-FLOW is unable to generate tsunamis formed from a multi-fault scenario, we therefore used COMCOT to propagate tsunami waves from layers 1 to 3. In the nested layer 4, we ran Delft3D-FLOW with boundary conditions based on water elevations produced in layer 3. We were able to vary Manning's roughness coefficient based on actual conditions, in reference to natural channels and flood plains (Arcement and Schneider, 1989).





The four simulation domain layers applied to COMCOT simulation all used spherical coordinates. Figure 3 shows the layers of the simulations used in COMCOT. Layer 4 was later converted to the modeling domain for use in the Delft3D simulations. Coordinates, grid size, and types of Shallow Water Equation (SWE) used in each individual layer are shown in Table 1. Delft3D-FLOW used the finest COMCOT layer as a domain, with coordinate transformation from spherical to Cartesian. We combined GEBCO and other nautical charts to produce improved bathymetry data. At the inland regions, we combined Shuttle Radar Topography Mission (SRTM) 30 with some cross topography measurements to improve the topographical data.

To simulate the rupture area caused by the $M_w$ 9.1 earthquake on December 26, 2004, we used a model proposed by Piatanesi and Lorito (2007) and Romano (2009). The rupture that generated the initial waves around the source area is shown in Fig. 4. We divided the rupture area into eight segments as described in Piatanesi and Lorito (2007). This model simulated a multi-fault scenario and was validated by Syamsidik et al. (2015).

We initiated sediment transport using a grain diameter switch for each of the three sites. By producing shear stress, we calculated a uniform Manning's roughness coefficient of 0.02 for the seafloor and a coefficient of 0.04 for overland areas combined with a water level fluctuation boundary condition from COMCOT simulations. We used Delft3D to simulate sediment transport processes caused by tsunami waves. We modeled this at the innermost layer of the simulation domain (layer 4 in the COMCOT model). Delft3D uses NSWEs combined with finite difference methods. The NSWEs in the model applies the equations of conservation mass, momentum, and energy flux and has the capability to simulate rapidly varying flow conditions (Stelling and Duijmeijer, 2003). Delft3D computes sediment transport using bedload and suspended load characteristics as described in van Rijn (1993; 2007). Changes in topography caused by erosion and sedimentation processes are updated at each time step of the simulation (Lesser et al., 2004). A more detailed explanation of the sediment transport formulas used in Delft3D can be found in Apotsos et al. (2011a). Recently, researchers have used Delft3D to simulate and test extreme events that triggered morphological changes, such as in the case of the Indian Ocean tsunami (Syamsidik et al., 2017; Gelfenbaum et al., 2007; Apotsos et al., 2011a,b,c). Previously, Delft3D was used and validated based on conditions less extreme than a tsunami, such as wind-wave-driven sediment transport (Broekema et al., 2016; Lesser et al., 2004; van Rijn et al., 2011).

### 3.2. Field Measurements

The results of the numerical simulations present estimations of the locations of the tsunami deposits, as well as their respective thicknesses. To assess the validity of our results, we conducted a series of surveys at three locations, i.e., Birek, Pasie Janeng, and Saney. We conducted these surveys in October 2015, January 2016, and April 2016 for Birek, Saney, and



Pasie Janeng, respectively. Surveys were conducted during different periods of the year because of the intensive labor and time needed for each survey. At each location, we performed manually excavated pit tests.

To define the tsunami deposit, we followed the USGS procedure as described in Peters and Jaffe (2010). The same

procedure was also applied during a tsunami deposit survey in Jantang conducted by Peters and Jaffe (2010). The identification of tsunami deposits relies on 10 criteria. Tsunami deposits are easily identifiable by their sharpness and erosional basal contact with original soil below or under the tsunami deposit. In this study, tsunami sediments were easily identified because of the characteristics of adjacent sediments, which have high organic contents from surface run-off processes. Tsunami sediments form thin layers, ranging between 1 and 30 cm thick in most cases (Srinivasalu et al., 2009).

Beaches near the study area are composed of sandy-quartz materials. Patches of coral reefs exist around Birek, Saney, and Pasie Janeng. Therefore, it was difficult to distinguish between tsunami deposits and sediments produced by littoral transport, also highlighted in Jaffe et al. (2003). Rip-up clasts were also found in the layers of the tsunami deposit layers, indicative of the energy that transported the material, such as in the case of a tsunami. Therefore, the existence of rip-up clasts further confirms the presence of tsunami deposits (Morton et al., 2007).

**4 Tsunami Run-Up and Tsunami Deposits**

We analyzed the impacts that the 2004 tsunami had on coastal morphology using field surveys and satellite images. Tsunami inundated areas were observed using satellite images to measure changes in coastal morphology in the vicinity of Lhoong, Aceh Besar. We identified tsunami deposits during a field survey in October 2015. Surveys started with coastal topography

measurements and then continued by locating potential deposits based on topography and surrounding conditions.

**4.1. Tsunami Inundation Area**

We analyzed the inundation area caused by the tsunami using satellite images captured in February 2005 by Digital Globe. After the 2004 tsunami propagated around the rupture source, waves moved inland, reaching the foothills approximately 200

m from the coastline at Birek and approximately 1 km inland at Jantang. The extent of inundation is shown in Fig. 5 by comparing the boundaries from numerical simulations and satellite images. The inundation limits generated by numerical simulation and those shown in the digitized images are generally consistent. On the basis of the inundation zones, we selected three locations to perform pit tests to sample the tsunami sediment deposits. For Jantang, we refer to descriptions in Jaffe et al. (2006).





## 4.2. Coastal Topography

Surrounded by hills, the topography conditions at the Birek coastal area are classified as V-shaped, indicating that the elevation of the center region is lower than its surroundings. At Birek, a small creek channels surface run-off to the sea.

Figure 6 shows the topography of Birek and cross profiles of the two transects, i.e., Cross 1 and Cross 2. The locations of the pit tests for the tsunami deposit sample collected are marked as B01, B02, and B03 in Fig. 6.

Pasie Janeng has a limited plain area, stretching approximately 100 m from the shoreline. Steep slopes characterize the inundation zone further inland at Pasie Janeng (Fig. 7). Given this topography, we estimate that surface run-off from rainfall,

flowing from the hillside to the beach, significantly affects the current thickness of tsunami deposits. We assume that hydrologic run-off processes are the main mechanism that erodes the tsunami deposits. At the Saney headland (Fig. 8), topography is flat, but the study area is sandwiched by the coastline. We conducted pit tests to investigate tsunami deposits at the Point S01 to Point S05 locations.

## 4.3. Sediment Characteristics

Tsunami deposits were identified by the sharp, clear presence of a sandy layer between new and old topsoil. The presence of seashells also strengthened the indication of tsunami deposits (Jaffe and Peters, 2010). Sediment deposition from the tsunami

waves almost reaches the foothills. The energy produced by the earthquake was capable of generating tsunami waves that deposited sediments up to 6.3 m in elevation above sea level. The thicknesses of the tsunami deposits correlate with distance from the shoreline, becoming thinner the farther inland. At sampling location B01, we did not find tsunami deposits. At location B02, tsunami deposit thickness was approximately 29 cm. The location of Point B02 is approximately 320 m from the coastline. At sampling location B03, we identified a thinner tsunami deposit, approximately 7.5 cm thick. This location is

approximately 400 m from the coastline (Fig. 9). Pit dimensions at these two locations were approximately 1.8 m in length, 0.9 m in width, and 1.50 m in depth.

The profiles of the pit tests at Pasie Janeng and Saney are shown in Fig. 10 and Fig. 11, respectively. Both Pasie Janeng and Birek have similar trends in thicknesses of tsunami deposits.

The majority of the sediment deposits at several locations in Lhoong contain sand, i.e., >70% sand (Birek = 82.15%, Pasie Janeng = 78.85%, and Saney = 71.88%). Waves carried the majority of the sand from the ocean bed, which mixed with deposits from each locality. Each sampling location contains less than 25% coarse grains (Birek = 6.3%, Pasie Janeng =

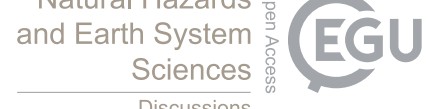

17.29%, and Saney = 20.46%), which is produced by local erosion. The tsunami flow depth drove the sediment transport process, which resulted in the characteristic sandy tsunami deposits. Additionally, we found rip-up clasts in the tsunami deposits at Birek and Pasie Janeng.

## 4.4. Model Hindcast

To simulate both hydrodynamic and morphodynamic processes during the 2004 Indian Ocean tsunami, COMCOT and Delft3D were run simultaneously. Transient waves produced by COMCOT successfully drove the sediment transport simulation performed by Delft3D. Tsunami wave heights, shear stresses, and the sediment transport process produced by Delft3D-FLOW are useful when trying to explain the sediment transport mechanism during the run-up of tsunami waves. Here, we present the results of the sediment transport process simulations in cumulative sedimentation and erosion thicknesses. Extreme changes in coastal morphology from plains to steep slopes significantly contributed to the wide distribution of tsunami deposits throughout the area affected by the tsunami.

### 4.4.1. Tsunami Wave Height and Sedimentation

Tsunami occurrence induced tsunami waves, inundating areas far inland. These results came first from COMCOT, which produced the tsunami wave height that was used as a parameter for Delft3D simulations. The modeled tsunami consisted of two major waves: (1) the first wave peaked at a height of 5.23 m and (2) the second wave peaked at a height of approximately 5.61 m (Fig. 12).

On the basis of the COMCOT results, Delft3D-FLOW, at a nearshore observation point, produced two major tsunami waves: (1) the first wave peaked at 12.2 m and (2) the second wave followed 14 min later peaking at 8.9 m, 3.3 m lower than the first wave. The fluctuation in run-up and backwash generated significant sediment transport in coastal waters and overland flow.

Sedimentation processes occurred predominantly during backwash. Specifically, such a case only occurs on relatively mild slopes. On the other hand, steep slopes remove sediments during the backwash. Reduced energy after the second and subsequent waves carried more sediment deposits, but had significantly less energy during the backwash process, therefore leaving more sediments behind (08:45 local time). As shown in Fig. 13, backwash produced a sediment deposit that was 0.38 m thick during the second wave.



### 4.4.2. Tsunami Velocity Fields

The sediment transport process is highly influenced by the shear process induced by tsunami currents along the bottom of the bed. The velocity fields during the tsunami run-up process help in explaining the energy involved during sediment transport processes. Figure 14 shows the velocity fields at the Birek and Pasie Janeng locations.

As shown in Figs. 14(a) and 14(b), during the wave's advance, the velocity at Birek location is lower than that when the wave is retreating. We also found the same condition at Pasie Janeng (Figs. 14(c) and 14(d)). The maximum value of shear stress was 83 N/m$^2$, recorded 24 min after the first wave arrived, eroding 0.11 m of the bed layer. High shear stress values significantly erode land, still depositing sediments 42 min after the first waves arrive.

### 4.4.3. Tsunami Deposit Thickness

On the basis of the simulations using both Delft3D and field measurements, we compare the differences in tsunami deposit thicknesses between the four sample locations. The run-up process mostly produces erosion, whereas most backwash does the opposite. The accumulation sedimentation map gave relatively reliable information on tsunami deposit locations, similar to the simulation results found for Lhoong. The following points compare the results of the numerical simulations and the
field data from the four sample locations.

a. Birek

Hydrodynamic process analysis revealed that there is some wave amplification, along with a decrease in the current velocity.
Elevated topography halted the tsunami run-up at the study area. The elevated topography deflected tsunami waves and reduced their velocity. The reduced tsunami flow velocity also signifies that the tsunami waves are losing energy. This explains the presence of a thinner tsunami deposit, which we found around the foothills compared with areas farther from the hills. The deflected waves created a larger wave around the hill, but at the same time, there was a significant decrease in the shear stresses. The lower the value of the shear stress that is generated by the waves, the more likely it is for sediment
deposition to take place. Figure 15 compares the results of the numerical simulation and field data at Birek.

The coastline is marked by the origin along the x-axis. Data from the three pit tests show that the numerical results overestimate the field data. It is important to note that this location has a V-shaped topography, surrounded by hills. The deposit thickness decreases with increasing distance from the coastline.



#### b. Pasie Janeng

Compared with Birek, we found slightly different results at Pasie Janeng (see Fig. 16). Areas with mild slopes are shorter than those at Birek, limiting the area capable of trapping sediments. The steep topography of the hills surrounding Pasie Janeng funnels rainfall run-off into a small channel that drains directly to the sea. The path that run-off took eroded a small

amount of the tsunami deposit that could have been deposited there for almost 12 years. The morphology of the site resembles an embayment, which potentially receives new sediment from coastal processes, making it difficult to find tsunami deposits nearshore.

#### c. Jantang

Jaffe et al. (2006) examined the thickness of tsunami deposits at Jantang during a survey conducted in 2005. The wide area, with mild slope, was advantageous for capturing sediments during the tsunami event. The relatively short time period between the tsunami and the survey was beneficial given that there was little activity in the area (i.e., not until 2009) just after the 2004 tsunami, concerning rehabilitation and reconstruction. Transect data based on field surveys were compared to

the numerical simulations. Comparisons between the field data and the numerical simulation results are shown in Fig. 17.

The results from the numerical simulations and the field survey are in good agreement. Unlike the data from the other three locations, the results of the numerical simulations at Jantang underestimate the field data. The absence of variations in the roughness coefficient slightly affected sediment transport results. The lack of additional land cover parameters from corals or

roads also affected several of the model calculated deposition and erosion zones. Besides these limitations, the simulation results are in good agreement with the field data.

#### d. Saney

Saney is protected by a small hill facing the ocean situated at the edge of the land, which made it more vulnerable to tsunami wave energy. A settlement complex was completely swept away during the 2004 tsunami. The tsunami wave that came from the west struck the edge of the land and amplified after encountering the mild slope area. The narrow headland of the site provided small place for the tsunami deposit to settling down. Years after the tsunami, other factors may have eroded the deposit, leaving thinner tsunami deposit. The results of the numerical simulations overestimated values compared with the

field data (see Fig. 18). Regardless witht the process, we still could found the tsunami deposit after 13 years of the event by the support of the numerical modeling.




## 5 Discussion

Estimations of the deposit locations and the thickness of tsunami deposits are key factors for site selection and target fieldwork efforts. Albeit 13 years after the Indian Ocean tsunami, numerical simulations still provide useful estimates in

order to locate tsunami deposits. Finding suitable tsunami deposit locations may become increasingly difficult as massive reconstruction efforts significantly alter the tsunami deposits in certain areas. The tsunami deposits may ultimately disappear or be difficult to identify. It is important to discuss land use practices with the local community in order to target well-preserved field sites.

We encountered the following limitations during the course of this study. First, field measurements could not provide the

exact number of tsunami waves. On the basis of our models, two main waves occurred during the tsunami in this region. This is corroborated by several eyewitness accounts from the study area given the absence of instrumental wave recordings. The small number of waves determined the size of the material contained in the deposits, such as the rip-up clasts in the tsunami deposit layer. We found rip-up clasts at both Pasie Janeng and Birek. This confirms that there were only a few waves, as this could be differed from storms waves (Morton et al., 2007). Secondly, there are uncertainties about the model

itself. For sediment transport, we used the 2DH Delft3D that did not account for turbulence kinetic energy, which may simplify the results. Furthermore, our simulations used depth-averaged velocity produced by the tsunami waves. This may, in fact, limit the maximum suspended sediment concentration because of the exclusion of suspended-sediment-induced density stratification (Jaffe et al., 2016). Other studies have attempted to run Delft3D with the 3-Dimensional model that allows interlayer settling and erosion processes during tsunami wave propagation (Apotsos et al., 2011b; Gelfenbaum et al.,

2007). Our study area has a complex topography and morphology. All of the sites, except for Jantang, have headlands and patches of coral reefs. The complexity of the topography could produce a significant difference between numerical model results and field data, as observed in the case of Sanriku coast in Japan (Goto et al., 2017). The roughness coefficient, which was set to a value similar to all grids of the sea area, could contribute to the different thicknesses found between the numerical simulations and the field data.

Regardless of the limitations and uncertainties, this study has successfully demonstrated the effectiveness of simulations in order to estimate the locations of tsunami deposits. Although natural processes such as rainfall, run-off, and aeolian sediment transport have taken place in the study area for more than a decade, location estimation produced by the numerical simulations is correct. The methods presented here are applicable to areas without any human activity or other extreme

events, such as effects of storms, that could not be considered in the simulations. Discontinuous and patchy tsunami deposit records may be the result of anthropogenic disturbances (Chague et al., 2017).



Further advances in the field of tsunami-induced sediment transport should incorporate collaborations between multi-disciplinary researchers, as this may increase the comprehensiveness of the study. The limitations and uncertainties highlighted in this study remain major challenges in paleo-tsunami studies (Sugawara et al., 2014a). A solid understanding of paleo-tsunamis, by exhibiting tsunami deposits to coastal communities, serves to allow us to better communicate the risk of

tsunamis to them.

## 6 Conclusions

This study has successfully demonstrated the coupling method of COMCOT and Delft3D-FLOW to provide a better understanding of tsunami-wave-induced sediment transport and to find the locations of tsunami deposits in a specific area more than one decade after the tsunami. Large shear stresses generated by tsunami waves transported sediment farther

inland. Field survey topography analysis also contributed to increased accuracy in defining the deposit location inside the tsunami inundation zone. Combining both methods contributed to results that are in good agreement between numerical simulations and field data. Tsunami deposit surveys were conducted at three locations between 2015 and 2016, showing that actual tsunami deposit thickness are thinner than the numerical results. Surface run-off, soil consolidation, and aeolian sediment transport could contribute to these differences. On the other hand, tsunami deposit data, obtained just after the

tsunami, provided a good fit to the numerical results. This study demonstrates the importance of rapid data collection in the immediate aftermath of a tsunami. Delft3D results displayed the location of sediment deposition, and field surveys confirmed the presence of tsunami deposits physically. Ultimately, the characterization of modern tsunami deposits will allow us to interpret earlier extreme events in sedimentary records.

**Acknowledgments**

We are grateful for support by Lhoong residents Mr. Alfaisal and Syahrul in the field survey. Data collection for this research is supported by PEER Cycle 5 Grant USAID and National Academy of Sciences funding this publication process through PEER Cycle 5, Sponsor Grant Award No. AID-OAAA-A-11-00012 and Subgrant No. PGA-2000004893 with research project titled: Incorporating Climate Change Induced Sea Level Rise Information into Coastal Cities' Preparedness

toward Coastal Hazards. Writing process of this paper has been part of World Class Professor Program 2017 (WCP 2017) Promoted by Ministry of Research, Technology, and Higher Education (RISTEKDIKTI) where Syiah Kuala University, Gadjah Mada University and Diponegoro University work in collaboration with Prof. Hermann M. Fritz.





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





Table 1. COMCOT simulation parameters.

| Layer | Longitude (º) | Latitude (º) | Grid size (m) | Type of SWE | Coordinate System |
|-------|---------------|--------------|---------------|-------------|-------------------|
| 1 | 79.2 to 107.6 | -13.6 to 8.2 | 1856 | Linear | Spherical |
| 2 | 94.61 to 97.78 | 3.41 to 6.29 | 618 | Linear | Spherical |
| 3 | 94.8 to 95.99 | 4.903 to 5.89 | 124 | Linear | Spherical |
| 4 | 95.15 to 95.31 | 5.25 to 5.33 | 18 | Non-linear | Cartesian |



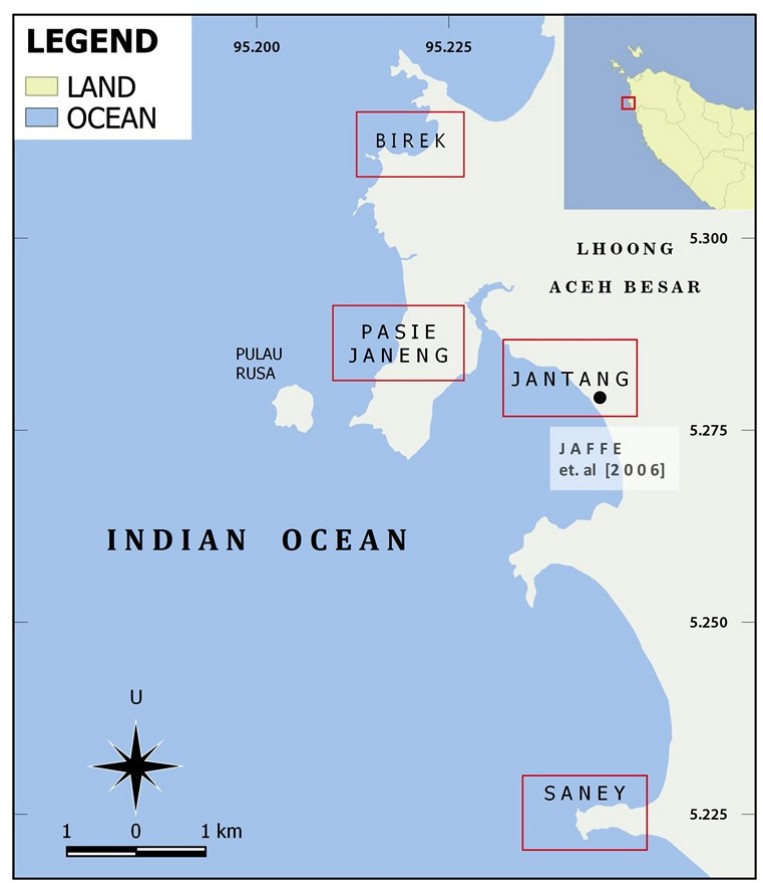

**Fig. 1:** Field survey sites at Lhoong, Aceh Besar, Sumatra, approximately 40 km south of Banda Aceh, Indonesia.





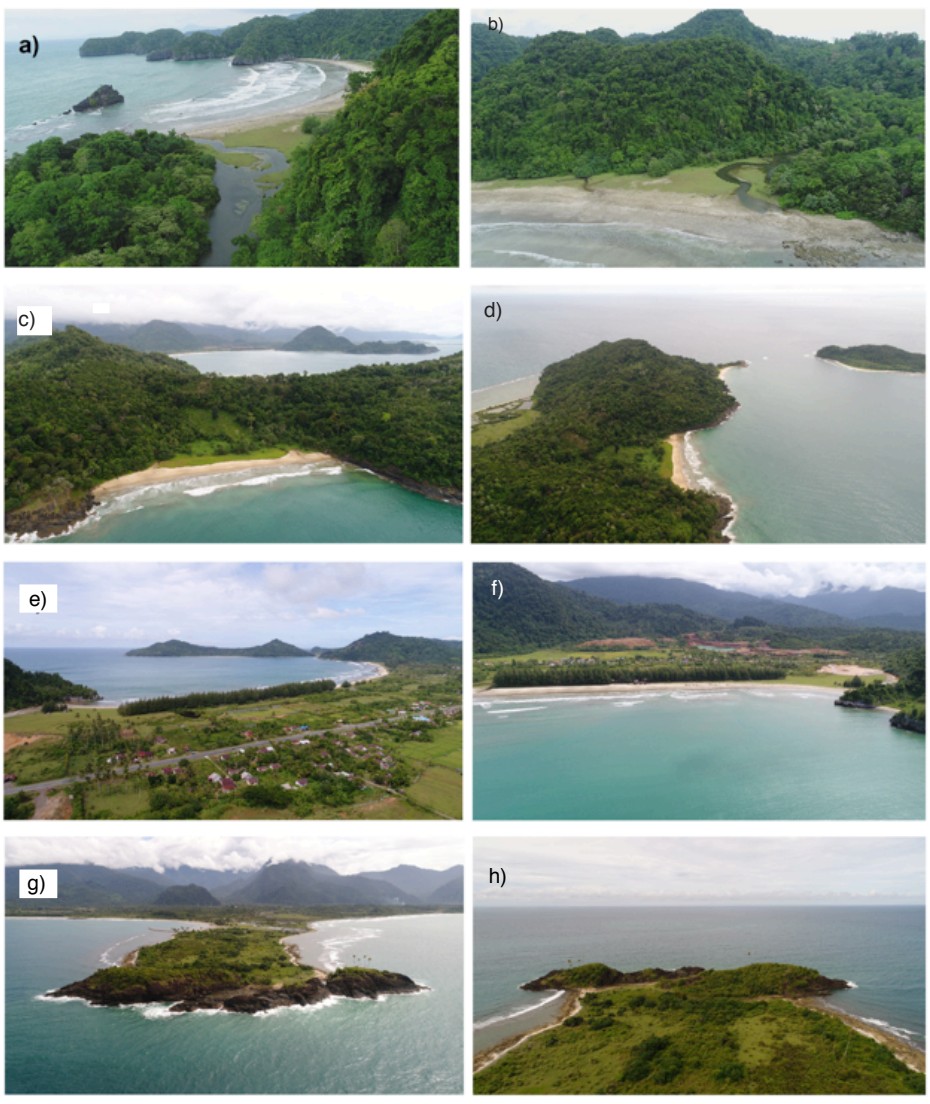

**Figure 2:** Aerial-drone photos of the study areas at Birek (a and b), Pasie Janeng (c and d), Jantang (e and f), and Saney (g and h).





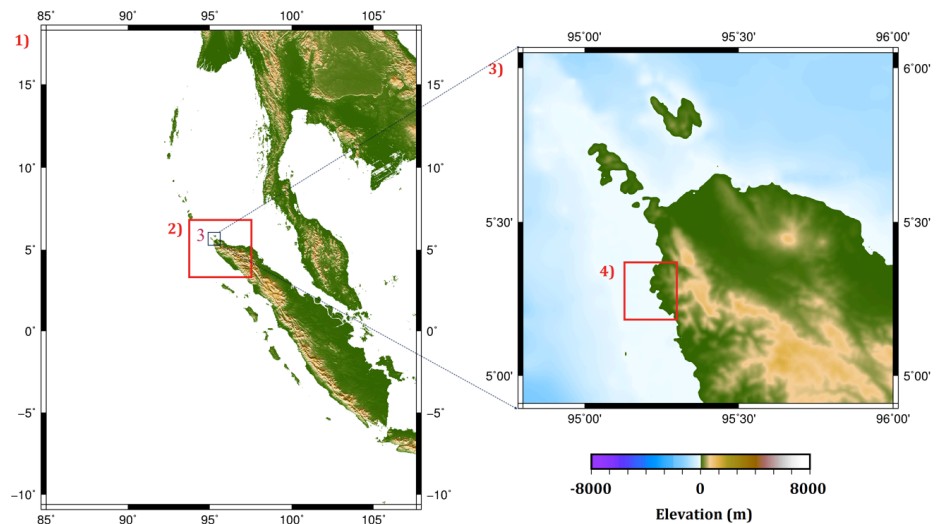

**Figure 3:** Nested simulation layers incorporated into COMCOT.





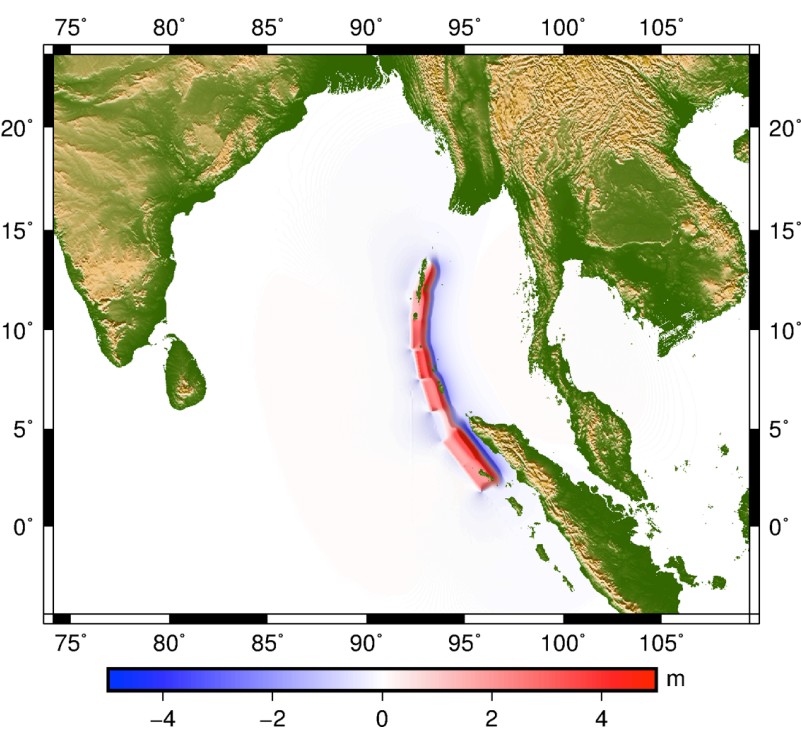

**Figure 4:** Initial wave of the fault scenario described in Piatanesi and Lorito (2007).



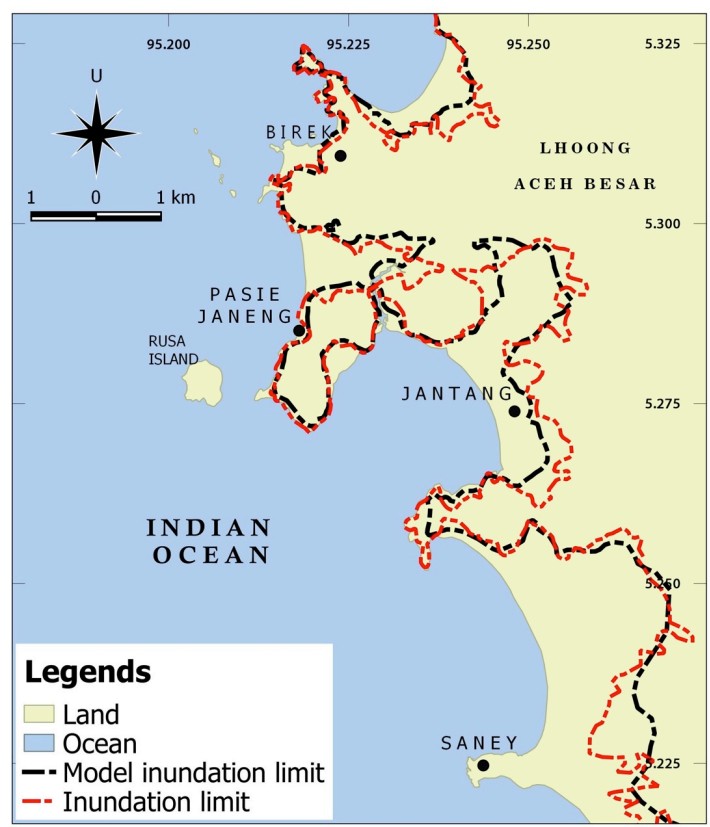

**Figure 5:** The 2004 tsunami inundation limit at Lhoong based on numerical simulation (black dashed lines) and satellite image digitization (red dashed lines).





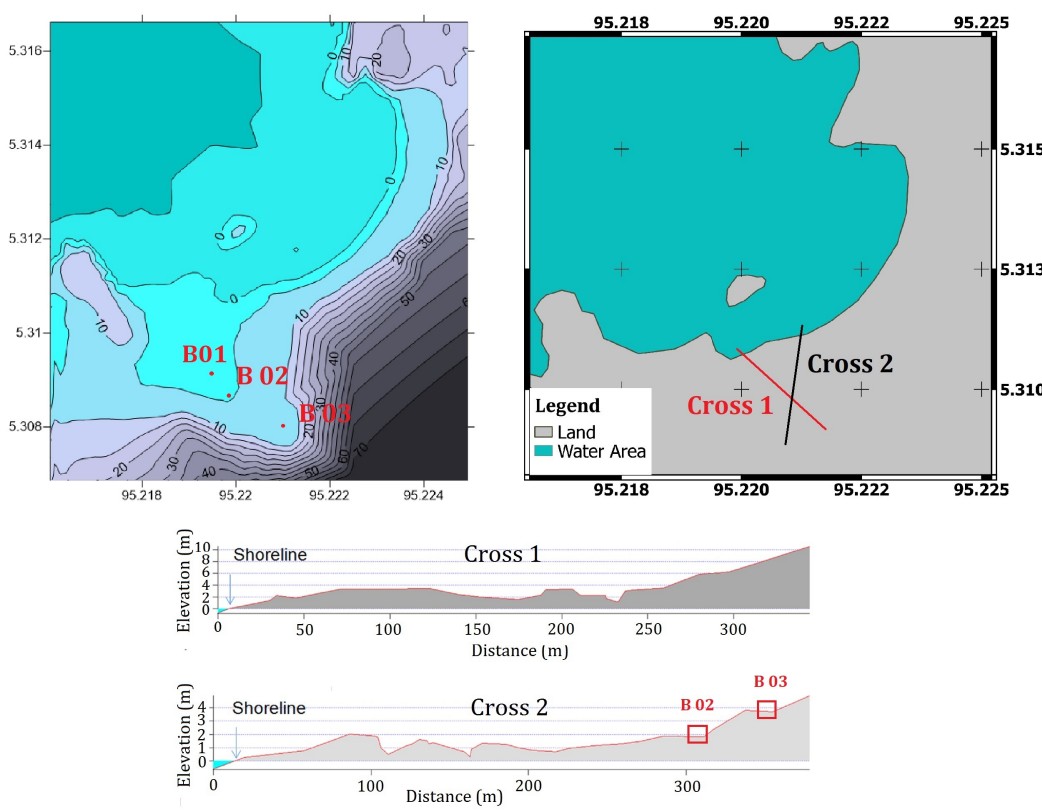

**Figure 6:** The topographic profile at Birek and the locations of the pit tests (B01, B02, and B03).





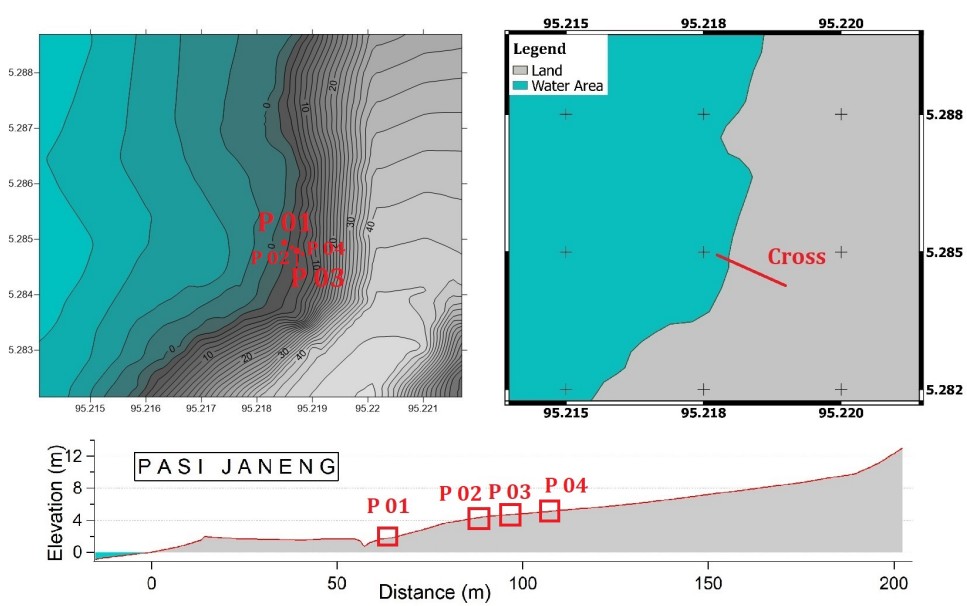

**Figure 7:** The topographic profile at Pasie Janeng and the locations of the pit tests (P01–P04).

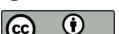


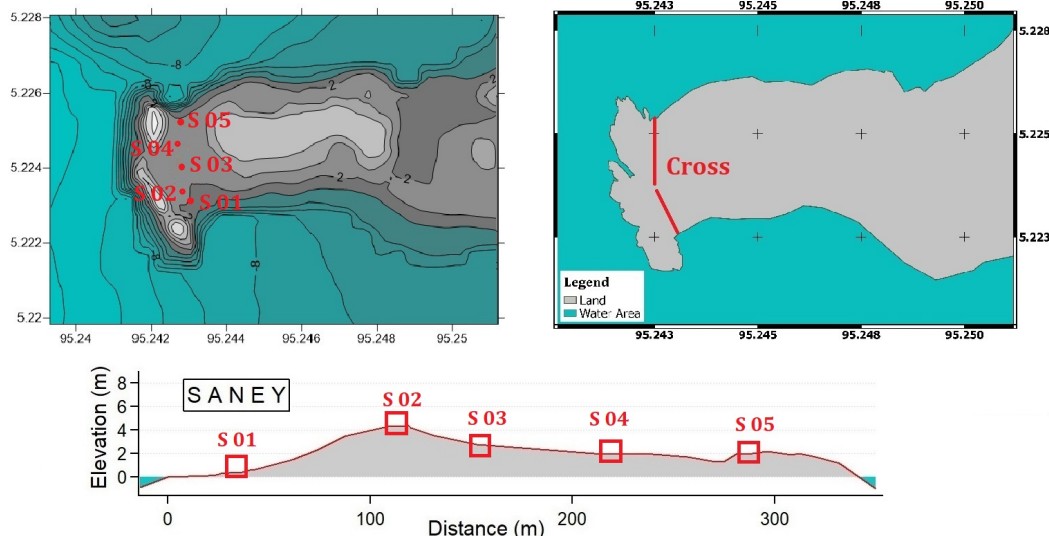

**Figure 8:** The topographic profile at Saney and the locations of the pit tests (S01–S05).





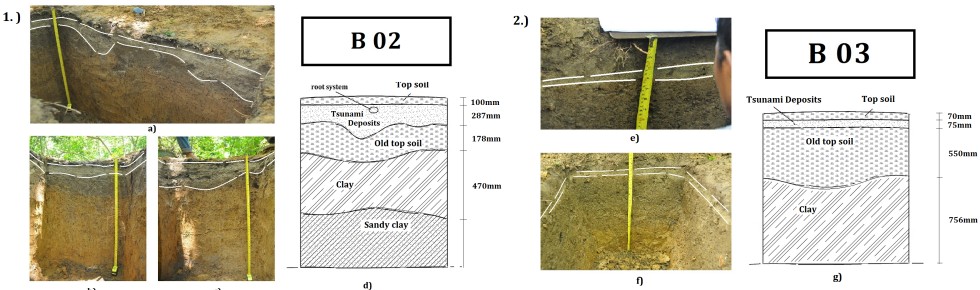

**Figure 9:** Profiles of tsunami deposit samples B02 and B03 at Birek.





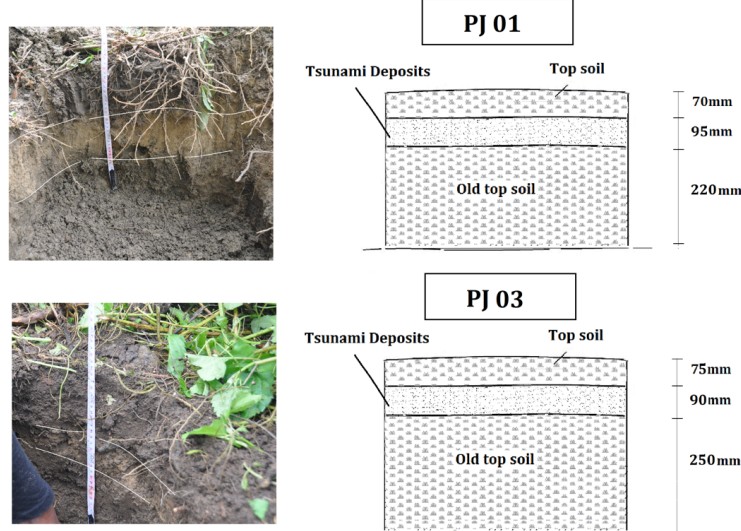

**Figure 10:** Profiles of tsunami deposit samples PJ 01 and PJ 03 at Pasie Janeng.





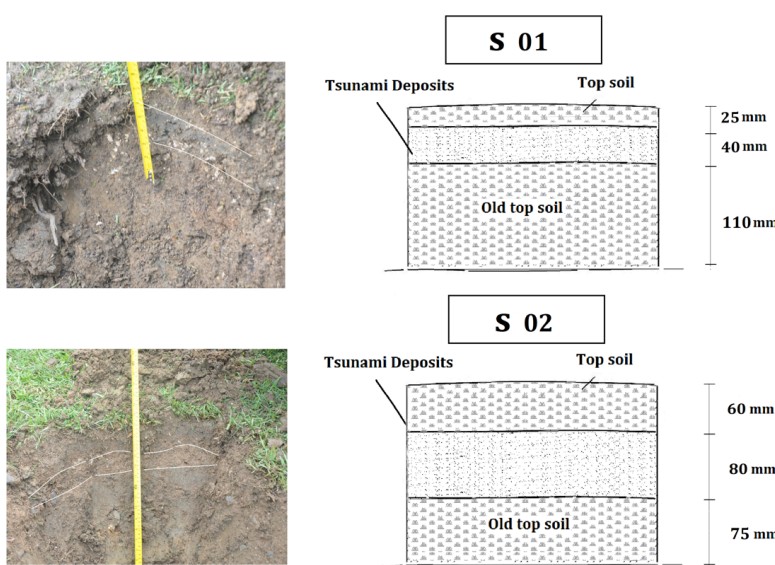

**Figure 11:** Profiles of tsunami deposit samples S01 and S02 at Saney.



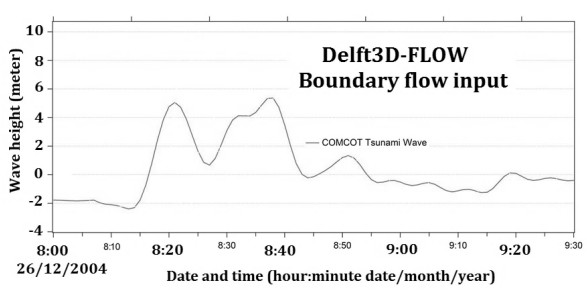

**Figure 12:** Delft3D-FLOW boundary flow input from COMCOT results.


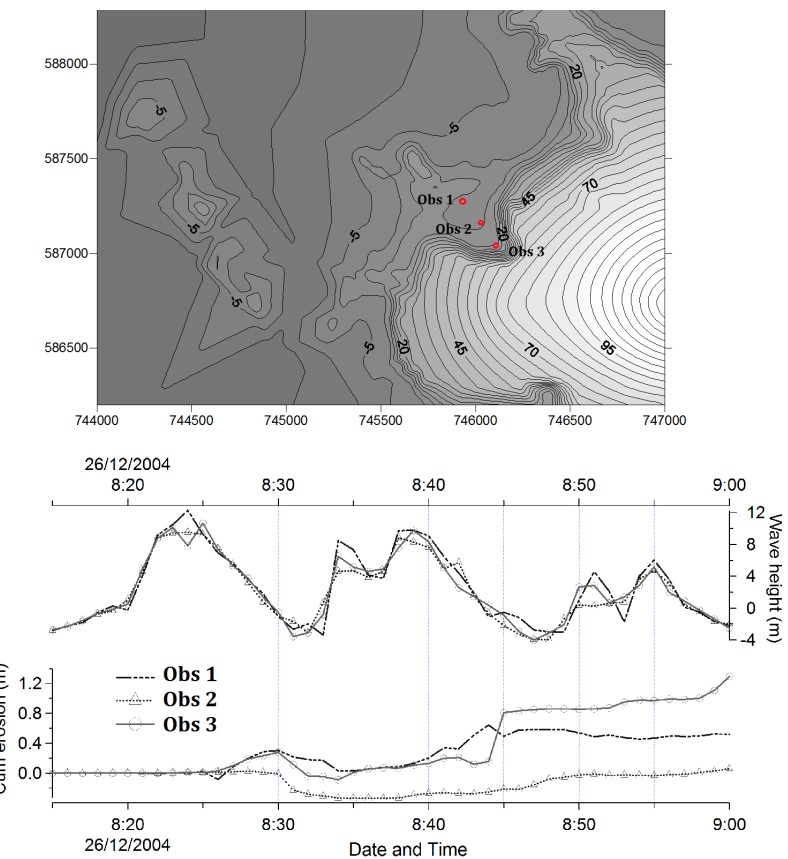

**Figure 13:** Correlation between wave height and cumulative sediment on the observation points.





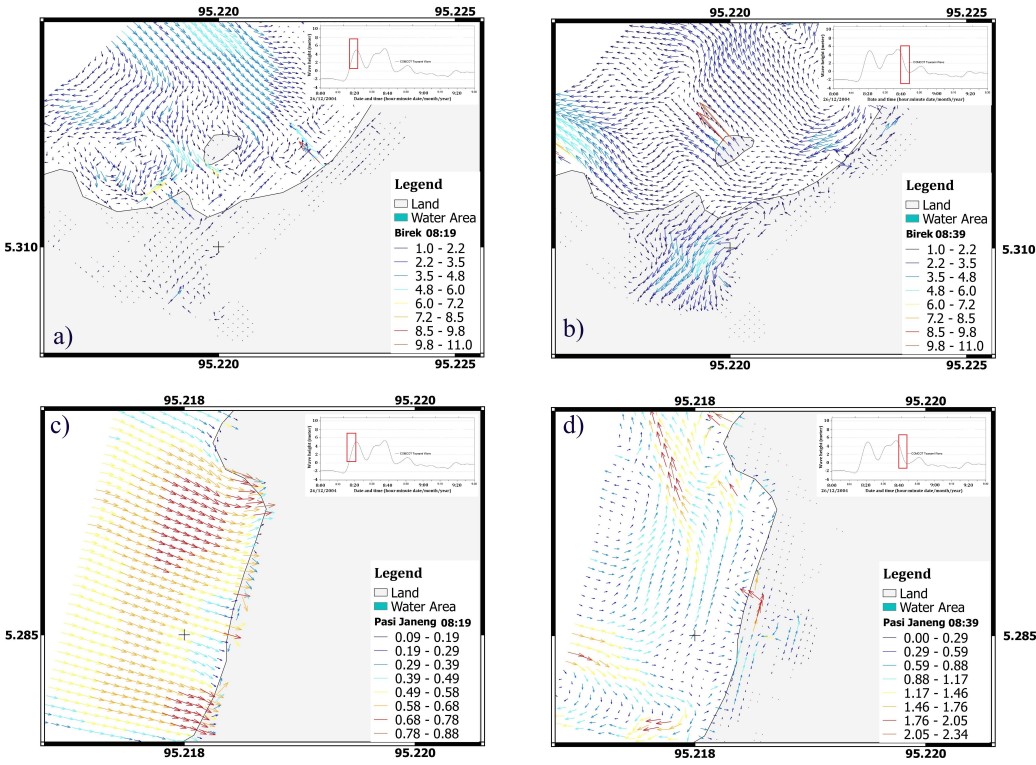

**Figure 14:** Tsunami wave velocity fields at Birek during wave advance (a) and during wave retreat (b) and at Pasie Janeng during wave advance (c) and during wave retreat (d).





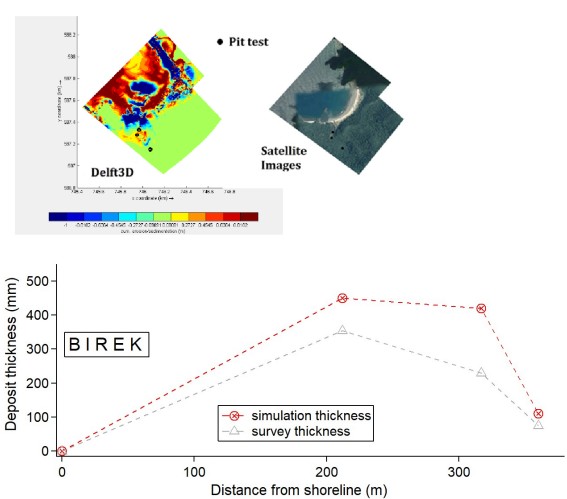

**Figure 15:** Tsunami deposit thickness for both the numerical model and the field survey at Birek.





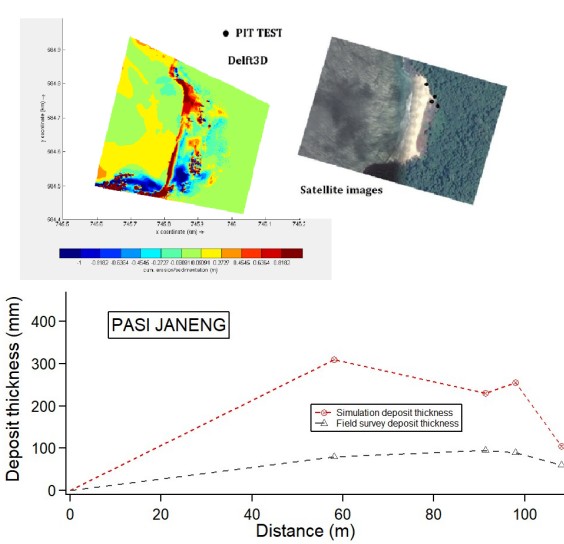

**Figure 16:** Tsunami deposit thickness results from the numerical model and the field survey at Pasie Janeng.





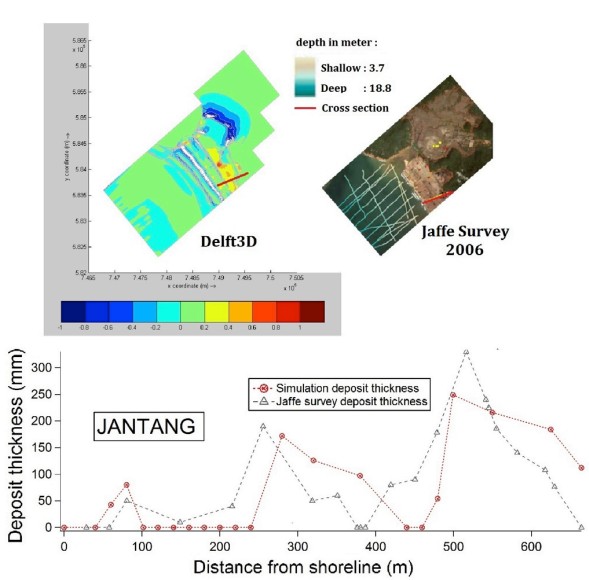

**Figure 17:** Tsunami deposit thickness results from the numerical model and the field survey at Jantang.

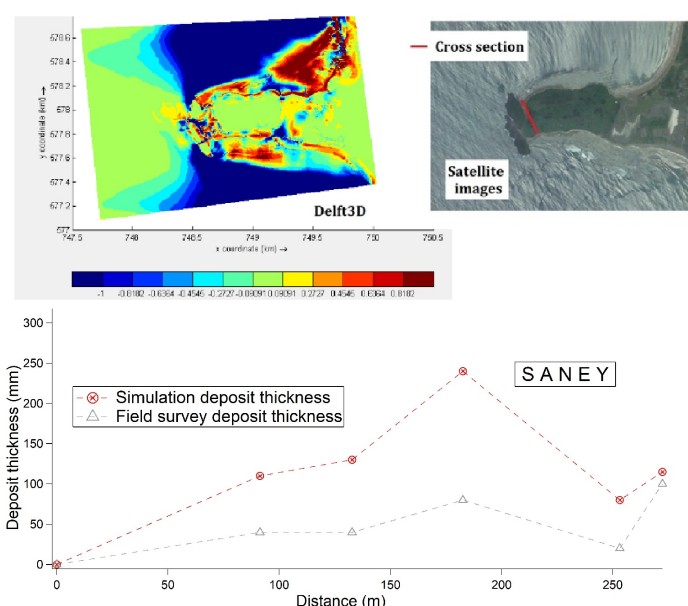

**Figure 18:** Results from the tsunami deposit thickness numerical model and field survey at Saney.