# Peer review of "Numerical Simulations of the 2004 Indian Ocean Tsunami Deposits Thicknesses and Emplacements"

_Natural Hazards and Earth System Sciences, 2018_

## Referee Comment (RC1) · Anonymous Referee #1 · 10 Dec 2018

Dear Editor, many thanks for the opportunity to review this manuscript on "Numerical Simulations of the 2004 Indian Ocean Tsunami Deposits Thicknesses and Emplacements" by Syamsidik et al.

Dear authors, I read with pleasure your manuscript focusing on coupling field and numerical data in an Indonesian region affected by the 26th of December tsunami.

Your manuscript is well-written and is easy to follow. However, I suggest that you largely reduce the number of images. Some are redundant while others can be easily merged (e.g. 3 and 4; 6, 7 and 8; 9, 10 and 11; 15, 16, 17 and 18).

Regarding the literature review there are several very important papers that are not

mentioned in the manuscript and need to be added (Paris et al., 2007; 2008; 2009; Costa et al., 2012; Szczuciński, 2012; among others). These papers discuss crucial aspects such as inundation phases, tsunami sediment sources and paths, geomorphological constrains, preservation issues and the authors will certainly benefit for reading these manuscripts. Some of their reasoning is questioned by these papers (for example, number of waves or inundation limit) and the authors need to acknowledge this and explain it.

For example, when the authors discuss post-depositional poor preservation, they need to understand and explain the natural processes behind it and clearly described in Szczuciński (2012). Moreover, when the authors mention that only two waves occurred in this region, they should discuss this in relation with the 7 waves described in nearby Lhok Nga (see papers by Paris mentioned above).

Furthermore, the geological criteria to identify tsunami deposits is very poorly described (e.g. "presence of sea shells") and some images are not clear enough to see the lithostratigraphical contrast (e.g. 11). To ascribe a deposit to a tsunami event you need many other criteria and you should clearly express that in the manuscript.

Finally, when you mention "As shown in Fig. 13, backwash produced a sediment deposit that was 0.38 m thick during the second wave." how did you confirmed that in the field? Costa et al. (2012) differentiated inundation and backwash with the shape of zircons, rounded and euhedral. Can you discuss this?

Your results are interesting and move tsunami geoscience forward. Coupling COMCOT with Delft 3D is interesting but you simply accepted the sediment transport formulas by default. You accepted Van Rijn formulas 1997 and 2007, why did you not test other formulas (please see Delft 3D FLOW Manual for many examples). Apotsos and Gelfenbaum work applied Delft 3D in a very specific context in American Samoa. The formula tests and results these authors obtained are obviously related with a context. You need to do the same and test, at least, other sediment transport formulas provided

by Delft 3D-FLOW.

Again, this is an interesting manuscript despite its weaknesses in sedimentological aspects and the straightforward application of a very competent open-source software.

In my opinion, this manuscript requires major changes before it is accepted for publication on NHESS. As mentioned above, the science is there but the authors need to redo some figures, add references, test new sediment transport formulas and totally reconsider its discussion based on previous findings described in the papers mentioned above.

Kind regards

———————————————

---

## Referee Comment (RC2) · Anonymous Referee #2 · 26 Dec 2018

This study investigated deposits by the Indian Ocean tsunami in 2004 thorugh the field measurement and reproduced it by using numerical simulations, which yields very interesting and valuable findings. However, the significance of this research should be more emphasized in the introduction. In the present manuscript, only the applicability of numerical models seem to be the main subject. The authors should show idea on how to make use of the results of this research in disaster prevention and reduction, such as estimating the magnitude of past tsunamis from sediments and evaluating the energy that the tsunami transports earth and sand. Although the authors stated that "this study estimated the energy required to transport sediments via a tsunami wave", the answer to this question is not clearly shown in the manuscript.

---

## Referee Comment (RC3) · Anonymous Referee #3 · 28 Dec 2018

This paper presents a phenomenon of tsunami-induced sediment transport, in particular for Aceh Besar, to develop a modeling technique from coupled numerical models. The technique provides a reliable and accurate examination of tsunami deposits for location and thickness. The topic is interesting for publication and the method is high quality. However, there is insufficient explanation to clarify the method. In addition, it seems that the amount of discussion should be increased according to the standard of this journal. Generally, the English wording and grammar need some improvement. After the revision and improvement, I feel that this paper would be suitable for publication in Natural Hazards and Earth System Sciences.

[Figure]

I have read this paper considering with the comments from other reviewers. In my opinion, this paper does not provide enough important information. I feel that the author is trying to hide or omit some theoretical background. I encourage the author to provide more explanation in each section including limitation of this numerical study. This is my initial comments and I will post my detailed comments later.

---

## Referee Comment (RC4) · Anonymous Referee #4 · 15 Jan 2019

Thank you for the efforts to understand the process of sediment accumulation by the tsunami a decade ago. The following are more detailed comment for the improvement of the paper

1. Fig. 15-18 are not readable

2. How to treat the open boundary conditions in D3D-flow model. Fig. 12 shows the boundary input for D3D-flow. Where this time series is obtained? How the others look like? Riemann or Neumann type conditions applied?

3. Please elaborate the data used for the simulation. Page 6, line 6: .. and other nautical charts? How details? Data resolution? This is very important also to be

included in the conclusion due the fact that the geometry of the beach is quite complex, especially for the smallest model domain.

4. Page 9, line 20. Where are these observation points located? Figure 13 does not show the location.

5. Still in page 9, please define the steep and mild slopes discussed in this paper. In fig. 13, why the obs no. 2 and 3, after 09.00, the sediment accumulation keep increasing?

6. Fig. 5: I think the authors would like to show readers that the points (observation or survey locations) should be within/inside the inundations limits. Point for Birek is outside the limits.

7. As it was discussed in page 2, it would be better also to produce and discuss extent of sediment deposition from the coastline (spatial distribution). The discussion on this topic is more meaningful and can also be used for disaster risk reduction related issues.

8. Please consider in the conclusion about the quality of the bathymetry/topography data that significantly influence the model results. This means there are so many weaknesses in this paper.

---

## Referee Comment (RC5) · Anonymous Referee #3 · 20 Jan 2019

The selected four areas in this study are not interfered by human, which is a reasonable criteria after more than a decade. However, the author mentioned that several prior studies have been conducted in the general study area. Apart from these four areas, is there any other conserved areas which can represent obvious coastal features, such as plain and ria? What did you find out during the area survey before the beginning of this study?

For numerical simulation, what if the author use only Delft3D for both hydrodynamic and morphodynamic models? I don't think that the multi-fault scenario is a reasonable reason. In addition, why don't the author perform COMCOT with NSWEs for layers 1

to 3? Please clarify more on this point. What is a novelty in this paper for numerical modeling?

The author verified tsunami inundation area from numerical simulations with satellite images. Please provide more explanation and clarification for the verification results.

Where is the location of boundary flow input in Figure 12?

I feel that the discussion part should be limitation of this study. Please clarify this.

In the conclusion part, the author mentioned that this coupling method of COMCOT and Delft3D provide a better understanding. How can the author conclude this? The method would be able to repeat sediment transport in these study areas. Lastly, what is the main benefit or contribution of this study?

---

## Author Comment (AC1) · 8 Mar 2019

COMMENTS FROM REFEREE #2: This study investigated deposits by the Indian Ocean tsunami in 2004 through the field measurement and reproduced it by using numerical simulations, which yields very interesting and valuable findings. However, the significance of this research should be more emphasized in the introduction. In the present manuscript, only the applicability of numerical models seem to be the main subject. The authors should show idea on how to make use of the results of this research in disaster prevention and reduction, such as estimating the magnitude of past tsunamis from sediments and evaluating the energy that the tsunami transports

earth and sand. Although the authors stated that "this study estimated the energy required to transport sediments via a tsunami wave", the answer to this question is not clearly shown in the manuscript.

RESPONSE:

Dear Referee #2,

We sincerely appreciate your input to our paper. In recent years, interests on using tsunami deposits information to estimate source of the tsunami and its magnitude have been in significant increase. Our study offers the use of numerical modeling prior to field investigation. This would provide a complimentary knowledge to locate potential locations of the tsunami deposits. In some studies, such as Jankaew et al. (2008) and Monecke et al. (2008), their important tsunami deposit findings were located after several attempts and identification of geological setting of the locations. Significance of our research is the use of the 2004 Indian Ocean tsunami with two different times of investigations, i.e one in 2005 (Jaffe et al., 2006 ) and in 2015 by authors. Understanding the energy of the tsunami from sediment transport process could be useful to validate energy generated by the tsunami. As in the 2004 Indian Ocean tsunami case, where monitoring equipments were rarely deployed in the affected area, information deduced from the tsunami deposit could better explain number of waves, wave heights, and wave velocity. In section 4.4, we discuss tsunami wave's shear stress simulated using the 2DH Delft3D model. The shear stress in sediment transport formulae represents the energy produced by the waves/currents during the process. We agree to include more explanation in the introduction part to elaborate the use of the study for disaster prevention and mitigation. Previous studies on tsunami deposits have inspired tsunami mitigation efforts in US and in Indonesia (Dunbar et al., 2008; Rubin et al., 2017). Data on tsunami deposits would provide more scientific evidence to past tsunamis for a significant period of records such as in the case of coastal cave in Aceh that preserved a very long record of past tsunamis in the region (Rubin et al., 2017). A set of tsunami deposits data could help us to estimate the recurrence period of the

tsunami from the subduction zone (Fujiwara, 2007; Minoura et al., 2001). Finding more tsunami deposits would strengthen validation process and ease some ways to estimate its sources through inverse mechanism (Buckley et al., 2012; Moore et al., 2011). A similar explanation will be added to the Introduction part of our revised manuscript.

References

Buckley, M.L., Wei, Y., Jaffe, B.E., Watt, S.G.: Inverse modeling of velocities and inferred cause of overwash that emplaced inland fields of boulders at Anegada, British Virgin Islands. Natural Hazards 63(1), 133-149. DOI: https://doi.org/10.1007/s11069-011-9725-8, 2012.

Dunbar P.K., Stroker, K.J., Brocko, V.R., Varner, J.D., McLean, S.J., Taylor, L.A., Eakins, B.W., Carignan, K.S., Warnken, R.R.:Long-Term Tsunami Data Archive Supports Tsunami Forecast, Warning, Research, and Mitigation. In: Cummins P.R., Satake K., Kong L.S.L. (eds) Tsunami Science Four Years after the 2004 Indian Ocean Tsunami. Pageoph Topical Volumes. Birkhäuser Basel, 2008.

Fujiwara, O.: Major contribution of Tsunami deposit studies to Quaternary Research. The Quaternary Research 46(3), 293-302. DOI: https://doi.org/10.4116/jaqua.46.293, 2007.

Jankaew, K., Atwater, B.F., Sawai, Y., Choowong, M., Charoentitirat, T., Martin M.E., and Prendergast A.: Medieval forewarning of the 2004 Indian Ocean tsunami in Thailand. Nature 455, 1228-1231, 2008. Minoura, K., Imamura, F., Sugawara, D., Kono, Y. and Iwashita, T.: The 869 Jogan tsunami deposit and recurrence interval of large-scale tsunami on the Pacific coast of northeast Japan. J. Nat. Disast. Sci. 23, 83–88, 2001.

Monecke, K., Finger, W., Klarer, D., Kongko, W., McAdoo, B.G., Moore A.L., and Sudrajat S.U.: A 1,000-year sediment record of tsunami recurrence in Northern Sumatra. Nature 455, 1232-1234, 2008.

Moore, A., Goff, J., McAdoo, B.G., Fritz, H.M., Gusman, A., Kalligeris, N., Kalsum, K.,

[Figure]

Susanto, A., Suteja, D., Synolakis, C.E.: Sedimentary Deposits from the 17 July 2006 Western Java Tsunami, Indonesia: Use of Grain Size Analyses to Assess Tsunami Flow Depth, Speed, and Traction Carpet Characteristics. Pure and Applied Geophysics 168(11), 1951-1961. DOI: https://doi.org/10.1007/s00024-011-0280-8, 2011.

Rubin, C.M, Horton, B.P, Sieh, K., Pilarcyk, J.E., Daly, P., Ismail N., and Parnell A.C.: Highly variable recurrence of tsunamis in the 7,400 years before the 2004 Indian Ocean tsunami. Nature Communications 8, 16019, 2017.

---

## Author Comment (AC2) · 8 Mar 2019

Dear Referee #4,

Thank you very much for your time to review our paper. Your scientific input to our paper have driven us to modify and revise our paper in order to ensure our research has been performed rigor and solid process. Our research was performed to provide answers to a complex system in determining locations of tsunami deposit prior to field investigation. We agree with your comments on this and we have been working at our best to address all your inputs in our revised manuscript. After carefully reading your comments, it took some time for us to revise and modify our paper as suggested. Now,

permit us to offer our response to your valuable comments.

COMMENT 1:

Thank you for the efforts to understand the process of sediment accumulation by the tsunami a decade ago. The following are more detailed comment for the improvement of the paper 1. Fig. 15-18 are not readable

RESPONSE 1: Thank you for your appreciation on our research. Fig. 15-18 have been modified and increased their resolutions to make it more readable. Combining Comments we received from other Referees, we have joined Fig. 15-18 and place them into one figure. Please see Fig. 1 of this response. The figure will replace Figs. 15-18. Furthermore, after being suggested to run more models, we have simulated the sediment transport models in DELFT3D using Engelund-Hansen 1967, Meyer-Peter-Mueller 1948, and Soulsby 1997. The original result presented in our previous manuscript was based on van Rijn 1984. Complete explanation of the mathematical formulae of the sediment transport can be find at Delft Hydraulic (2009). More detailed explanation of the additional simulation results for the sediment transport formulae can be seen at our Responses to Referee #1.

COMMENT 2:

2. How to treat the open boundary conditions in D3D-flow model. Fig. 12 shows the boundary input for D3D-flow. Where this time series is obtained? How the others look like? Riemann or Neumann type conditions applied?

RESPONSE 2: The Open Boundary condition here was placed about 14.5 km from shoreline. The hydrodynamic boundary condition was obtained from COMCOT numerical simulation at Layer #3. Here, the boundary condition was set to follow "Water Level". The water level boundary condition is a modified Riemann boundary condition where Stelling has added the time-derivative to water level and and velocities (Stelling, 1984). It was meant to reduce reflection process caused by the eigen frequency of the

simulation.

COMMENT 3: 3. Please elaborate the data used for the simulation. Page 6, line 6: .. and other nautical charts? How details? Data resolution? This is very important also to be included in the conclusion due the fact that the geometry of the beach is quite complex, especially for the smallest model domain.

RESPONSE 3: Yes, we certainly agree that the resolution of the data used in the simulation is important. Bathymetry data for Layers 1-3 were adopted from GEBCO data, with 0.5 minutes resolution. Meanwhile bathymetry data for Layer 4 were re-digitized from Nautical Charts released by Indonesian Navy (PUSHIDROSAL) with Map No. 15-2014. Topography data for Layer 4, where all of the sediment deposition and erosion were mapped, were adopted from topography map released by Indonesia Agency for Geospatial (BIG) interpreted from DEM data with resolution of 5 m. Some area around the transects sampling areas were also measured using handheld GPS, water pass, and staffs. The measured data were also corrected to local tide data. Similar explanation will be added to Subsection3.2 Field Measurement.

COMMENT 4: 4. Page 9, line 20. Where are these observation points located? Figure 13 does not show the location.

RESPONSE 4: Please see three red circles in upper figure of Fig. 13, showing Obs 1, Obs 2, and Obs 3. Obs 1 is the closest observation point to the coastline. Meanwhile Obs 4 is the farthest point from the coastline and it is just before the wall of the hill.

COMMENT 5: 5. Still in page 9, please define the steep and mild slopes discussed in this paper. In fig. 13, why the obs no. 2 and 3, after 09.00, the sediment accumulation keep increasing?

RESPONSE 5:

Here, we define the slope larger than 0.02 is steep and smaller than that was classified as mild. The topography condition at Birek can be seen at upper left part of Fig. 2 (part

of figures in this response) showing topography contour. As it can be inferred from Fig. 2 of this response or Fig. 13 in the previous version of our manuscript, the dense contour lines represent the steep slope of the area. Here, the slope of the area about 0.015 for the first 250 m from coastline (mild). Suddenly, after the point, the slope were between 0.05 until 0.1 (steep). The land area slope at Pasie Janeng was about 0.05 (steep). At Saney the slope of the land area was about 0.013 (mild). At Obs 2 and Obs 3, the sedimentation accumulation keep increasing due to weakening of shear stress generated by the tsunami waves after the second waves. The decrease of the shear stress drove the sediment to settle at the points. Similar explanation will be added into 4.4 Model Hindcast in our revised manuscript.

COMMENT 6: 6. Fig. 5: I think the authors would like to show readers that the points (observation or survey locations) should be within/inside the inundations limits. Point for Birek is outside the limits.

RESPONSE 6: Yes, thank you for your suggestions. We have modified Fig. 5 by moving all the points inside the inundation limits. Originally the points represented the administratif location of the each villages, which pointed the location of the Office of the Village Administration. Since here we do not deal with the administrative issue, we agree to follow the suggestion as can be seen in the Attachment (see Fig. 3 of this response).

COMMENT 7: 7. As it was discussed in page 2, it would be better also to produce and discuss extent of sediment deposition from the coastline (spatial distribution). The discussion on this topic is more meaningful and can also be used for disaster risk reduction related issues.

RESPONSE 7: Erosion areas (negative sedimentation/erosion accumulation as in Fig. 2 of this response) were located at nearshore area where energy of the tsunami waves was still large. This implies that any structures or objects that could reduce shear stress by increasing roughness coefficient could help to mitigate impacts of tsunami.

Here, the some of the higly eroded area did not have sufficient coastal vegetation that could increase manning coefficient. Deposition were largely found farther from coastline where energy of the tsunami wave decreased. Similar explanation will be added at Section 5 Discussion in our revised manuscript.

COMMENT 8: 8. Please consider in the conclusion about the quality of the bathymetry/topography data that significantly influence the model results.

RESPONSE 8: Thank you for your suggestions. We acknowledge the resolution of the bathymetry/topography data are crucial in the simulation. For pre-survey estimation of the location of the tsunami deposit, our simulation offers scientific basis to decide where sampling should be done. It is certain that detail bathymetry data, especiall at nearshore area, could increase the reliability of the estimation. Similar explanation will be added in at Section 6 Conclusion in our revised manuscript.

Again, allow us to reiterate our sincere appreciation for thoughts and time given by the referee to ensure our paper meet the standard of the journal. Thank you very much.

REFERENCES

Delft Hydraulics: Delft3D-FLOW Simulation of multi-dimensional flows and transport phenomena, including sediments. Delft, 2009.

Stelling, G.S.: On construction of computational methods for shallow water flow problems. Tech. Rep.. Delft, 1984.

[Figure]

**Legends**

○ Deposit thickness obs. point

Deposit thickness (m)
- ■ < -1.5meter
- ■ -1.5 - -1.3
- ■ -1.3 - -1.1
- ■ -1.1 - -0.9
- ■ -0.9 - -0.7
- ■ -0.7 - -0.5
- ■ -0.5 - -0.3
- ■ -0.3 - -0.1
- ■ -0.1 - 0.1
- ■ 0.1 - 0.3
- ■ 0.3 - 0.5
- ■ 0.5 - 0.7
- ■ 0.7 - 0.9
- ■ 0.9 - 1.1
- ■ 1.1 - 1.3
- ■ 1.3 - 1.5
- ■ > 1.5meter

**Fig. 1.** Accumulative sedimentation and erosion at Birek, Pasie Janeng, Jantang, and Saney. Negative values represent erosion and positive values represent sedimentation thicknesses.

[Figure]

**Fig. 2.** Topography condition of Birek, Pasie Janeng, and Saney.

**Fig. 3.** The 2004 tsunami inundation limit at Lhoong based on numerical simulation (black dashed lines) and satellite image digitization (red dashed lines).

---

## Author Comment (AC3) · 9 Mar 2019

Dear Referee #3,

We thank for your valuable input to our paper. Your comments have been highly considered in the revision process of our manuscript. Three more sediment transport formulas have been incorporated in our simulations that enable us to provide more insightful result and solid arguments on our research results. We appreciate that you have put your comments into two entries and we have incorporated them rigorously enhance the quality of our paper. Now, allow us to respond to the first entry of your comments in more detailed description.

[Figure]

COMMENT 1: This paper presents a phenomenon of tsunami-induced sediment transport, in particular for Aceh Besar, to develop a modeling technique from coupled numerical models. The technique provides a reliable and accurate examination of tsunami deposits for location and thickness. The topic is interesting for publication and the method is high quality.

RESPONSE 1: Thank you very much for your appreciation on our research. We humbly offer our efforts to increase our understanding on tsunami hazards, especially on using tsunami-induced sediment transport to reveal more scientific basis for field measurements and disaster mitigation as well.

COMMENT 2: There is insufficient explanation to clarify the method. In addition, it seems that the amount of discussion should be increased according to the standard of this journal.

RESPONSE 2: We appreciate your input to clarify our methods. Taking into accounts of your comments and other referees comments, we will add more explanation on

COMMENT 3: Generally, the English wording and grammar need some improvement. After the revision and improvement, I feel that this paper would be suitable for publication in Natural Hazards and Earth System Sciences.

RESPONSE 3: Thank you for your suggestion. Originally, this manuscript has undergone a professional proof reading process. We ensure that the quality of the english our this paper has been checked by native-english proof reader. Notwithstanding with the proof-read manuscript, we would be improving the wording and grammar of our revised manuscript.

COMMENT 4: I have read this paper considering with the comments from other reviewers. I feel that the author is trying to hide or omit some theoretical background. I encourage the author to provide more explanation in each section including limitation of this numerical study. This is my initial comments and I will post my detailed comments
later.

RESPONSE 4: Thank you for highlighting the important of the theoretical background of our study. Combining your comments with all the referees comments, we have added some explanation on data resolution used in the simulation, on manning roughness coefficient, limitation on bathymetry data used, three additional sediment transport formulae (i.e. Engelund-Hansen 1967, Meyer-Peter-Mueller 1948, and Soulsby 1997) in the simulations, analysis of the results from four sediment transport formulae, and explanation of topography condition of the simulation area. In the original form of our manuscript we presented the result based on van Rijn 1984 Sediment Transport formulae. Detailed documentation of the sediment transport formulae can be seen in Delft Hydraulics (2009). We also highlight the limitation of our study in the discussion section (Section 5) and in the Conclusion (Section 6) in our revised manuscript. Your detailed comments are highly appreciated and we would be addressing our further responses in the next entry of your Comment.

Thank you very much.

REFERENCE

Delft Hydraulics: Delft3D-FLOW Simulation of multi-dimensional flows and transport phenomena, including sediments. Delft, 2009.

---

## Author Comment (AC4) · 9 Mar 2019

Dear Referee #3,

We sincerely appreciate your extra effort to comment on our paper in two stages. These support our objective to disseminate our research result with scientific evidence and solid process of discussion. Your second stage comment is divided into eight comments followed by our response to each of your comment. Now, permit us to respond to your second stage of comments.

COMMENT 1:

[Figure]

The selected four areas in this study are not interfered by human, which is a reasonable criteria after more than a decade. The author mentioned that several prior studies have been conducted in the general study area. Apart from these four areas, is there any other conserved areas which can represent obvious coastal features, such as plain and ria?

RESPONSE 1: Thank you for confirming our reasons in selecting our study area. The area, Lhoong of Aceh Besar, has been part of the study area for paleotsunamis by Rubin et al. (2017) and Jaffe et al. (2006). The two studies have also motivated our team to select the study area. Data collected by Jaffe et al. (2006) were also used to compare our numerical simulations and their data. As Jaffet et al.'s data were collected after some months of the 2004 Indian Ocean tsunami, therefore, they could provide the closest data to the event without being significantly altered by other processses, either natural or anthropogenic processes. There is no significant area of Lhoong that can be categorized as ria coast. At the north of Saney, there is an area where it could be classified as plain coast with about 3 km long. However, this area has undergone a heavy anthropogenic intervention after the tsunami as it was paddy field and settlement area. Therefore, we determine the four study sites as the most tsunami deposit conserved area. Similar explanation could be found in our present manuscript in Page 3 Lines 15 – 32 (Section 2 Study Area).

COMMENT 2: What did you find out during the area survey before the beginning of this study?

RESPONSE 2: We interviewed some local people to understand whether any significant intevention has been made after the tsunami that could disturb the tsunami deposit. We confirm again this by conducting preliminary observation of the area to clarify the earlier information received from local people. The area where we did pit tests and trenches were largely deserted from any settlement and farming activities. This helped us to exclude the anthropogenic influence. On the other hand, surface run-off or other natural process could interfere the area.

COMMENT 3: For numerical simulation, what if the author use only Delft3D for both hydrodynamic and morphodynamic models? I don't think that the multi-fault scenario is a reasonable reason. In addition, why don't the author perform COMCOT with NSWEs for layers 1 to 3? Please clarify more on this point.

RESPONSE 3: The version of DELFT3D that we operate does not allow us to put the offshore multi-fault scenario that could reconstruct the 2004 Indian Ocean tsunami case. The multi-fault scenario for the 2004 Indian Ocean tsunami is more valid than the single fault (Piatanesi and Lorito, 2007; Koshimura et al., 2009 ; Syamsidik et al., 2015; Romano, 2009). All these study have validate the tsunami waves generated using tsunami poles (inland), measured water marks on buildings, and water level measured offshore by Satellite Jason 1. Therefore, we confirm that the use of multi-fault scenario of the 2004 Indian Ocean tsunami is reliable. Using NSWE for layers 1 to 3 provide insignificant different to the wave generated around the offshore area. On the other hand, applying non-liner SWE on Layers 1 and 3 will reduce computational time without overseeing the accuracy of the simulation results. Thank you for commenting on these. We will also add similar explanation in Section 3 (Methods) in our revised manuscript.

COMMENT 4: What is a novelty in this paper for numerical modeling?

RESPONSE 4: Coupling a numerical hydrodynamic module with a numerical sediment transport module in the case of tsunami wave studies is an important part of this study. In addition, reliablity of the study have been revealed using field data collected from four locations. We also demonstrate the use of the numerical simulation combined with the understanding on geological settings to locate the tsunami deposits. Similar explanation could be found in the Introduction part (Section 1). More statements will be added in the section to strengthen the novelty side of our research in our revised manuscript.

COMMENT 5: The author verified tsunami inundation area from numerical simulations with satellite images. Please provide more explanation and clarification for the verifica-

tion results.

RESPONSE 5: We redigitize the tsunami inundation area based on Landsat Satellite Images captured on February 2nd, 2005 or about 1 month after the disaster. This allowed us to clearly identify the extend of the damaging area based on the images and compared them with the tsunami inundation area produced by Delft3D-Flow. Please see the satellite image at Fig. 1 of this response.

COMMENT 6: Where is the location of boundary flow input in Figure 12? I feel that the discussion part should be limitation of this study. Please clarify this.

RESPONSE 6: The location of the boundary can be seen in Fig. 2 of this response where a blue line was drawn inside Layer 4. Later, Layer 4 was transformed into Delft3D-FLOW's simulation domain. We agree to further discuss in in Subsection 3.1 Numerical simulations. Limitation of the study will also be discussed in Section 5 (Discussion) in our revised manuscript.

COMMENT 7: In the conclusion part, the author mentioned that this coupling method of COMCOT and Delft3D provide a better understanding. How can the author conclude this? The method would be able to repeat sediment transport in these study areas.

RESPONSE 7: In the present form of COMCOT, it is unable to simulate sediment transport module. There have been several attempts to include the sediment transport modules in COMCOT (Li et al., 2012 ; Rasyif et al., 2019). However, they are in preliminary stage of the application. This research offers results where spatial distribution of the sedimentation and erosion with transient process can be investigated by coupling it with Delft3D-FLOW. Here, we could understand wheter backwash play significant role on the process and whether shear stress corellate to the accumulation of the sediment deposit transported by the tsunami waves.

COMMENT 8: Lastly, what is the main benefit or contribution of this study?

RESPONSE 8: This study contribute to explain the spatial distribution of sedimentation

and erosion generated by the 2004 Indian Ocean tsunami. Such study that uses the 2004 Indian Ocean tsunami by numerically reconstruct the sedimentation and erosion spatial distribution are rare (Li et al., 2012; Gusman et al., 2012). This study also offers the use of the numerical simulation coupled with understanding of geological setting of the study to estimate locations of the tsunami deposits after12 years of the event (field data collected in 2015 and 2016).

We would like to reiterate our gratitude to the Referee for commenting our research. The comments support us to leverage our study for having a better understanding on tsunami hazards. Thank you very much.

REFERENCES

Gusman, A.R., Tanioka, Y., and Takahashi, T.: Numerical experiment and a case study of sediment transport simulation of the 2004 Indian Ocean tsunami in Lhok Nga, Banda Aceh, Indonesia. Earth, Planets and Space 64, 3, 2012.

Jaffe, BE, Borrero, JC, Prasetya, GS, Peters, R, McAdoo, B, Gelfenbaum, G, Morton, R, Ruggiero, P, Higman, B., Dengler, L., and Hidayat, R.: Northwest Sumatra and offshore islands field survey after the December 2004 Indian Ocean Tsunami. Earthquake Spectra 22, 105-135, 2006.

Koshimura, S., Oie, T., Yanagisawa, H., and Imamura, F.: Developing fragility functions for tsunami damage estimation using numerical model and post-tsunami data from Banda Aceh, Indonesia, Coastal Engineering Journal, 51(3), 243-273, 2009.

Li, L., Qiu, Q., and Huang, Z.: Numerical modeling of the morphological change in Lhok Nga, west Banda Aceh, during the 2004 Indian Ocean tsunami: understanding tsunami deposits using a forward modeling method. Natural Hazards 64, 1549-1574, 2012.

Piatanesi, A. and Lorito, S.: Rupture process of the 2004 Sumatra- Andaman earthquake from tsunami waveform inversion. Bulletin of the Seismological Society of America 97, 223-231, 2007.

Rasyif, T. M., Kato, S., Syamsidik, and Okabe, T.: Numerical Simulation of Morphological Changes due to the 2004 Tsunami Wave around Banda Aceh City, North Sumatra, Indonesia. Geosciences, 2019 (accepted for publication).

Romano, F.: The rupture process of recent tsunamigenic earthquake by geophysical data inversion. Doctoral Thesis, Universita Degli, Bologna, Italy, 2009.

Rubin, C.M, Horton, B.P, Sieh, K., Pilarcyk, J.E., Daly, P., Ismail N., and Parnell A.C.: Highly variable recurrence of tsunamis in the 7,400 years before the 2004 Indian Ocean tsunami. Nature Communications 8, 16019, 2017.

Syamsidik, Rasyif, T.M., and Kato, S.: Development of accurate tsunami estimated times of arrival for tsunami prone cities in Aceh, Indonesia. International Journal of Disaster Risk Reduction 14, 403-410, 2015.

[Figure]

**Fig. 1.** Landsat satellite image captured on February 2nd, 2005 showing the tsunami inundation area (DigitalGlobe, 2019).

**Fig. 2.** Layers 3 and 4 of the COMCOT simulation domain. Blue line represents the open boundary location for Delft3D-FLOW. Red circle for the observation point for boundary.

---

## Author Comment (AC5) · 11 Mar 2019

Dear Referee #1:

We thank to your valuable comments and your time to review our manuscript. Your comments have driven some important changes on our paper, such as performing more numerical simulations by incorporating three other sediment transport formulae (Engelun-Hansen 1967, Meyer-Peter-Mueller 1948, and Soulsby 1997), combining some figures, and adding more explanation and discussion in our revised manuscript. The lack of spatial distribution of sediment transport studies driven by the 2004 Indian Ocean tsunami has been one of our motivations in this study. Beside this, there have

been a number of paleotsunami study conducted in the northern part of Sumatra that can also used to strengthen this research. Your comments are very much appreciated to ensure our manuscript meets scientific standard of the NHESS journal and useful for further development in tsunami sciences. Now, permit us to respond to your comments in more details. We divide your comments into 7 comments followed by our response to each of them.

COMMENT 1: Dear Editor, many thanks for the opportunity to review this manuscript on "Numerical Simulations of the 2004 Indian Ocean Tsunami Deposits Thicknesses and Emplacements" by Syamsidik et al. Dear authors, I read with pleasure your manuscript focusing on coupling field and numerical data in an Indonesian region affected by the 26th of December tsunami. Your manuscript is well-written and is easy to follow.

RESPONSE 1: We humbly offer our gratitude to your appreciation to our paper. We are pleased to learn that you acknowledge the manuscript is easy for you to follow and it has a well-written structure of a scientific manuscript. Thank you very much for this important statement.

COMMENT 2: I suggest that you largely reduce the number of images. Some are redundant while others can be easily merged (e.g. 3 and 4; 6, 7 and 8; 9, 10 and 11; 15, 16, 17 and 18).

RESPONSE 2: Thank you for suggesting to reduce some figures. Please find some figures attached in this response to demonstrate that we have combined some figures into one and will incorporate them into our revised manuscript. Merged Figures 3 and 4 is shown in Figure 1 of this response, combined Figures 6, 7, and 8 is shown in Figure 2 of this response, Figures 9,10, and 11 are merged become Figure 3 of this response, and Figures 15, 16, 17, and 18 are combined become Figure 4 of this response.

COMMENT 3: Regarding the literature review there are several very important papers that are not mentioned in the manuscript and need to be added (Paris et al., 2007;

2008; 2009; Costa et al., 2012; Szczucinski, 2012; among others). These papers discuss crucial ′ aspects such as inundation phases, tsunami sediment sources and paths, geomorphological constrains, preservation issues and the authors will certainly benefit for reading these manuscripts. Some of their reasoning is questioned by these papers (for example, number of waves or inundation limit) and the authors need to acknowledge this and explain it. For example, when the authors discuss post-depositional poor preservation, they need to understand and explain the natural processes behind it and clearly described in Szczucinski (2012). Moreover, when the authors mention that only two waves occurred ′ in this region, they should discuss this in relation with the 7 waves described in nearby Lhok Nga (see papers by Paris mentioned above).

RESPONSE 3: Thank you for suggesting importance and related references to be included in our manuscript. Paris et al. (2007) performed their study around Lhok Nga of Aceh Besar, which is about 20 km to the east of our study areas. Their study exhibited the influence of local topography on the sediment thicknesses found in the area. Thickest deposit was found at low topography situation and and steep slopes gave varied results in spatial distribution of the tsunami deposit. This was found true in Birek and Pasie Janeng of our study sites that are surrounded by hillsides. Costa et al. (2012; 2015) proposed the shape of the zircons in the sediment deposit could be used to interpret number of waves, and tsunami run-in or backwash processes. Szczucinski (2012) provided an excellent basis for our study, especially on consideration of any process followed after the tsunami that could erode or alter the tsunami deposit. Our study area is situated in a tropical area where rainy season occurs about 4-5 months in a year with high precipitation rate. After more than 10 years, the 2004 Indian Ocean tsunami deposit in this area have encountered some natural processes despite selection of the study area has made to select the most sediment preserved area in Aceh Besar district. Szczucinski also argues that tsunami inundation less than 3 m would unlikely to preserve the sediment deposit years after the tsunami event (Szczucinski, 2012). We are pleased to include them in Section 1 Introduction, Section 3 Methods, and Section 5 Discussion in our revised manuscript. Thank you very much for the suggestions.

COMMENT 4: Furthermore, the geological criteria to identify tsunami deposits is very poorly described (e.g. "presence of sea shells") and some images are not clear enough to see the lithostratigraphical contrast (e.g. 11). To ascribe a deposit to a tsunami event you need many other criteria and you should clearly express that in the manuscript.

RESPONSE 4: The presence of sea shells were identified through a microscopic observation of the sediment material. This could distinguish the tsunami-induced deposit from orignal top-soil material or other surface run-off process. Other criteria of the tsunami deposit were advised by Jaffee et al. (2003 ) and Peters and Jaffe (2010) where they put the methods of the tsunami-induced sediment transport investigation into one practical guideline. We followed exactly the steps of the guideline and clarify it using some microscopic observations. We spent a significant period (from 0.5-2.0 hours) for each of sample to carefully identify the tsunami deposit and distinguish it from other sources of sediment. In total, we collected 14 samples of the tsunami deposits and 22 locations of sampling performed by Jaffee et al. (2006). Simplified lithostratigraphical contrast of our surveys could be seen in Figure 3 of this response. We will add similar explanation to Section 3 Methods in our revised manuscript.

COMMENT 5: Finally, when you mention "As shown in Fig. 13, backwash produced a sediment deposit that was 0.38 m thick during the second wave." how did you confirmed that in the field? Costa et al. (2012) differentiated inundation and backwash with the shape of zircons, rounded and euhedral. Can you discuss this?

RESPONSE 5: We base our arguments on the wave heights from the simulation as presented in Figure 13 in our manuscript. Backwash process based on the shape of the sediment was beyond our field investigation. We appreciate the suggestion to refer to Costa et al. (2012) where the study was performed with detailed litostratigraphy processes, such as exoscopic, radio-carbon dating and micro palaenthology. The latter three processes were part of the limitation of our study. Costa et al. (2015) stated that euhedral zircon could be associated to backwash process. Meanwhile, rounded zircons could be attributed to deposition occurred during tsunami run-in process. The

detailed observation is absence in our present study. It is certain that the references could leverage our future field investigation on tsunami-induced sediment transport.

COMMENT 6: Your results are interesting and move tsunami geoscience forward. Coupling COMCOT with Delft 3D is interesting but you simply accepted the sediment transport formulas by default. You accepted Van Rijn formulas 1997 and 2007, why did you not test other formulas (please see Delft 3D FLOW Manual for many examples). Apotsos and Gelfenbaum work applied Delft 3D in a very specific context in American Samoa. The formula tests and results these authors obtained are obviously related with a context. You need to do the same and test, at least, other sediment transport formulas provided by Delft 3D-FLOW.

RESPONSE 6: At the beginning of our study, we have performed a number of literature reviews on a number of sediment transport formulae used to simulate tsunami-induce sediment transport. One study prooved that van Rijn formulas 1997 provided the best results compared to tsunami-induce sediment transport experiments in a wave flume (Li and Huang, 2013). Notwithstanding with the reviews we conducted, we agree to follow your suggestions to incorporate more sediment transport formulas, such as Engelund-Hansen 1967, Meyer-Peter-Mueller (MPM) 1948, and Soulsby 1997. One of the results of the simulation can be seen in Fig. 6 of this response to compare our simulation results to sediment deposit measured by Jaffe et al. (2006) at Jantang. Using linear regression method, we found the van Rijn 1993 formulas gave the best approximation to the field data. Similar results were found at other three study sites with smaller r square. Comparison from all simulated sediment thicknesses to field data at four sites can be seen at Figure 5 of this Response. Figure 5 will be included in our revised manuscript in Section 4.4.3 Tsunami Deposit Thicknesses. Discussion of the results will be also elaborated further in Section 5 Discussion.

COMMENT 7: Again, this is an interesting manuscript despite its weaknesses in sedimentological aspects and the straightforward application of a very competent open-source software. In my opinion, this manuscript requires major changes before it is

accepted for publication on NHESS. As mentioned above, the science is there but the authors need to redo some figures, add references, test new sediment transport formulas and totally reconsider its discussion based on previous findings described in the papers mentioned above. Kind regards

RESPONSE 7: Thank you for your comments. We are pleased to present our findings and follow comments from all referees, including your comments. Your comments are very valuable. They have driven a number of changes and motivated us to present clearer arguments and findings in our revised manuscript.

REFERENCES

Costa, P.J.M., Andrade, C., Cascalho, J., Dawson, A.G., Freitas, M.C., Paris, R., and Dawson, S.: Onshore tsunami transport mechanism inferred from heavy mineral assemblages. The Holocene 1-15, 2015. DOI:10.1177/0959683615569322.

Costa, P.J.M., Andrade, C., Freitas, M.C., Oliveira, M.A., Lopes, V., Dawson, A.G., Moreno, J., Fatela, F.F., and Jouanneau, J-M.: A tsunami record in the sedimentary archive of the central Algarve coast, Portugal: Characterizing sediment, reconstructing sources and inundation paths. The Holocene 1-16, 2012. DOI: 10.1177/0959683611434227.

Jaffe, B., Gelfenbaum, G., Rubin, D., Peters, R., Anima, R., Swensson, M., Olcese, D. Bernales L., Gomez, J., and Riega, P.: Tsunami Deposits: Identification and Interpretation of Tsunami Deposits from the June 23, 2001 Peru Tsunami, Proceedings of the International Conference on Coastal Sediments 2003, CD-ROM Published by World Scientific Publishing Corp and East Meets West Productions, Corpus Christi, TX, USA. ISBN 981-238-422-7, 13 p. 2003.

Jaffe, B.E, Borrero, J.C, Prasetya, G.S, Peters, R, McAdoo, B, Gelfenbaum, G, Morton, R, Ruggiero, P, Higman, B., Dengler, L., and Hidayat, R.: Northwest Sumatra and offshore islands field survey after the December 2004 Indian Ocean Tsunami. Earthquake Spectra 22, 105-135, 2006.

Li, L. and Huang, Z.: Modeling the change of beach profile under tsunami waves: A comparison of selected sediment transport models. J. Earthq. Tsunami 7, 2013, DOI:10.1142/S1793431113500012.

Paris, R., Lavigne, F., Wasmer, P., and Sartohadi, J.: Coastal sedimentation associated with the December 26, 2004 tsunami in Lhok Nga, west Banda Aceh (Sumatra, Indonesia). Marine Geology 238(1-4), 93-106, 2007. DOI: 10.1016/j.margeo.2006.12.009.

Peters, R. and Jaffe, B.E.: Identification of tsunami deposits in the geologic record: developing criteria using recent tsunami deposits. U.S. Geological Survey Open-File Report 2010-1239, 39 p., 2010.

Szczucinski, W.: The post-depositional changes of the onshore 2004 tsunami deposits on the Andaman Sea coast of Thailand. Natural Hazards 60, 115-133, 2012. DOI 10.1007/s11069-011-9956-8.

**Fig. 1.** Numerical Simulation Layers and Domain in COMCOT and Delft3D-FLOW (Layer 4 of COMCOT)

[Figure]

**Fig. 2.** Topography condition of Birek, Pasie Janeng, and Saney.

**Fig. 3.** Tsunami Deposit Features found at Pit Test Locations

**Legends**

○ Deposit thickness obs. point

**Deposit thickness (m)**

■ < -1.5meter
■ -1.5 - -1.3
■ -1.3 - -1.1
■ -1.1 - -0.9
■ -0.9 - -0.7
■ -0.7 - -0.5
■ -0.5 - -0.3
■ -0.3 - -0.1
■ -0.1 - 0.1
■ 0.1 - 0.3
■ 0.3 - 0.5
■ 0.5 - 0.7
■ 0.7 - 0.9
■ 0.9 - 1.1
■ 1.1 - 1.3
■ 1.3 - 1.5
■ > 1.5meter

**Fig. 4.** Spatial Distribution of Acumulation of Sedimentation and Erosion obtained from Numerical Simulations.

[Figure]

**Fig. 5.** Comparison of four sediment transport formulae applied in Delft3D-FLOW to Field Data

[Figure]

van Rijn 1993

y = 0.2276x
R² = 0.38594

MPM 1948

y = 0.093x
R² = -0.0088

Soulsby 1997

y = 0.1358x
R² = -0.1233

Engelund-Hansen 1967

y = 0.1866x
R² = 0.02893

**Fig. 6.** Regression result of sediment transport numerical simulation and field data collected in Jantang by Jaffee et al. (2006).